# Effects of Reduced Amino Acids and Apparent Metabolizable Energy on Meat Processing, Internal Organ Development, and Economic Returns of Cobb 700 and Ross 708 Broilers

**DOI:** 10.3390/ani15071064

**Published:** 2025-04-06

**Authors:** Bo Zhang, Shengyu Zhou, Wei Zhai, Yang Zhao

**Affiliations:** 1The Department of Poultry Science, Mississippi State University, Starkville, MS 39762, USA; bozhmsu@gmail.com (B.Z.); wei.zhai@msstate.edu (W.Z.); 2The Department of Animal Science, The University of Tennessee, Knoxville, TN 37996, USA; zhoushengyuxnkl@gmail.com

**Keywords:** amino acid, energy, broiler, processing, economic return

## Abstract

Broiler chickens have been bred for fast growth and high feed efficiency, but this has led to health and welfare challenges, including excessive body fat, muscle disorders like woody breast, and footpad dermatitis. These issues not only reduce meat quality but also lower economic returns for poultry producers. One possible solution is adjusting the balance of protein and energy in their feed. This study examined how reducing dietary amino acids and energy affects growth, organ development, meat quality, and economic value in two common broiler strains, Cobb 700 and Ross 708. The study found that reducing amino acids by 30% increased body fat while decreasing the incidence of severe woody breast and shortening the length of footpad dermatitis. Lowering dietary energy by 16% reduced fat accumulation and improved muscle yield and economic returns but slightly reduced the proportion of normal breast meat. Based on these findings, the study recommends specific protein-to-energy ratios for different growth stages to help farmers balance broiler health, meat quality, and profitability. These results provide practical feeding strategies to improve both broiler welfare and economic efficiency in poultry production.

## 1. Introduction

The production performance of broilers has been continuously improved over the past seven decades [1]. However, commercial broilers have exhibited many problems, including increased fat pad weight, woody breast myopathy (WBM), and footpad dermatitis [2,3,4].

The WBM, footpad dermatitis, and fat pad weight have been found to reduce economic return, the welfare of broilers, and carcass quality. The WBM has emerged as a new problem for the broiler industry since about 61.3% of commercial broilers exhibited medium and severe WBM at d 45 [5]. Although the consumption of WBM meat does not pose any health issues to consumers, it is linked with low water holding capacity and increased hardness to textural properties, which are all indicators of poor meat quality [6]. If the decreased meat qualities negatively affected the customers’ preferences, then WBM meat would be sold at a lower price [7]. The fast-growing birds were likely to develop footpad dermatitis, which would limit walking abilities and cause broilers’ welfare problems [4]. Furthermore, as commercial birds age and consume more dietary energy, they tend to accumulate larger quantities of abdominal fat, resulting in the storage of energy as less valuable fat and diminishing the potential for higher marginal returns [8,9]. Measures should be taken to mitigate WBM incidence, footpad dermatitis, and fat pad weights.

The above-mentioned problems are related to fast growth and high dietary density [3,4,8]. Dietary nutrient reduction is one of the methods to control the growth rate and reduce these problems. The WBM incidence decreased at 42 d of age when dietary lysine was reduced by 30% from 12 to 28 d of age [3]. Broilers’ footpad dermatitis on day 35 was reduced by dietary 3% protein reduction [10]. Although the fat pad yield could not be decreased by dietary amino acid (AA) reduction, dietary 10% energy reduction has been shown to control fat pad weight at 42 d of age [9,11]. However, the effects of the dietary reduction in both AA and energy on the fat pad weight, WBM incidence, and footpad dermatitis were not yet clear. 

The reduction in dietary AA and AME affects broiler growth by modulating blood metabolites related to protein and lipid metabolism. The reduction in dietary AME limits the broilers’ capacity to efficiently utilize excess protein, leading to its accumulation in the intestines. This not only results in nutrient wastage but also promotes the proliferation of harmful intestinal bacteria [12], ultimately impairing gut health. Conversely, dietary protein deficiency or an imbalanced AA profile can slow growth rates, increase fat deposition, and damage the intestinal mucosa and barrier function [13]. These impairments reduce nutrient digestion and absorption, thereby negatively impacting broiler performance. 

An excessively low protein-to-energy ratio results in disproportionately high energy intake relative to protein, which has been shown to significantly elevate plasma triglyceride concentrations in broilers [14,15]. This imbalance stimulates hepatic lipogenesis, as indicated by increased activities of key lipogenic enzymes, including fatty acid synthase, malate dehydrogenase, and acetyl-CoA carboxylase [16]. As a result, lipogenesis and fat deposition are enhanced in response to excess energy intake [17]. Furthermore, broiler growth is closely regulated by serum hormone levels. Broilers fed low-protein diets exhibit elevated plasma concentrations of growth hormone [18] and thyroid hormones [19], which together stimulate skeletal muscle development and protein synthesis [20]. These hormonal changes enhance the utilization efficiency of the limited dietary protein and reduce plasma uric acid levels, the end product of protein catabolism, thereby mitigating the physiological consequences of protein deficiency.

Dietary AA and energy reductions have been found to affect internal organ development in broiler birds. The whole intestine weights were decreased in broilers fed a 10% energy-reduced diet at 14 d of age [9]. However, our previous study also found that duodenum and jejunum relative weights at 60 d of age were increased when broilers were continuously fed a 20% AA-reduced diet from 0 to 60 d of age [21]. The differences in intestinal responses to dietary AA and energy decrease may be attributed to the age of broilers used in various research studies investigated. However, the effects of a continuous dietary reduction in both energy and AA on internal organ development were still not clear in broilers beyond age 6 weeks. Therefore, more studies are still needed.

Dietary nutrient reduction has also been suggested to decrease feed costs and carcass yields, influencing economic returns. Marginal returns for Ross 708 broilers increased when dietary digestible Lys was decreased from 1.80% to 1.02% from 0 to 25 d of age [22]. However, Zhai, et al. [23] found that broilers with methionine at 80% of the NRC-recommended level had the lowest feed cost, but the BW and carcass yield were severely reduced. Therefore, it is necessary to find out the optimal level of dietary nutrient reduction that could balance the feed cost and carcass yield. Moreover, WBM-affected meat would be sold at a lower price and decrease marginal returns so WBM incidence can be used as an indicator that may decrease economic returns. However, most previous studies failed to consider WBM incidence in economic analysis. Therefore, the current study will investigate the effects of different levels of dietary AA and energy reduction on economic return with consideration of WBM incidence. Additionally, this study will investigate the responses of internal organ development and carcass yield to dietary nutrient reductions. The current study aims to identify the optimal level of dietary nutrient reduction by examining the effects of reduced amino acids and energy on broiler processing, internal organ development, and economic returns until 9 weeks to improve profitability for broiler companies.

## 2. Materials and Methods

### 2.1. Birds, Diets, and Management

The experiment was conducted following the principles and specific guidelines of the Institutional Animal Care and Use Committee at Mississippi State University, IACUC Animal Welfare Assurance #A3160-01. The current study consisted of 2 trials with Cobb 700 and Ross 708, respectively, in trials 1 and 2. Both trials used a randomized complete block design with factorial arrangements of 12 treatments (4 AA × 3 AME). In the trial, each of 6 replicate blocks contained 12 pens with each pen randomly assigned to one of the dietary treatments. In each trial, 864-day-old broiler chicks were obtained from a commercial hatchery and were randomly distributed into 12 pens, with 6 male and 6 female broilers in each pen. The digestible AAs (lysine, total sulfur AA, and threonine) in 4 diets were 70%, 80%, 90%, and 100% of breeder recommendations for Cobb 700 and Ross 708 broilers [24,25]. The apparent metabolizable energy in 3 diets was 84%, 92%, and 100% of the breeder recommendation for Cobb 700 and Ross 708 broilers. All birds were fed in a 4-phase program (Starter (day 0–14, Table 1), Grower (day 14–28, Table 2), Finisher (day 28–41, Table 3), and Withdrawal (day 41–60, Table 4) grow-out period phases) for 9 weeks. For detailed information on the feed ingredients and nutrient composition across the 12 dietary treatments in all four phases, please refer to Appendix A. More detailed information about bird usage, diet formulation, and management was available in the companion study [26]. 

### 2.2. Woody Breast Myopathy Evaluation 

At 33, 40, and 47 d of age, all birds were manually evaluated for woody breast myopathy conditions in both trials. Skinless breasts of broilers were also scored for WBM severity after deboning at 62 and 64 d of age. The method of WBM condition evaluation was modified from previous studies [3,27]. Manual palpation was used to evaluate the WBM severity based on a scale from 0 to 3. Score 0 represented a normal breast in which no firmness was detected in the whole breast. Score 1 represented a slight WBM that firmness was detected in the cranial region of the breast. Score 2 represented a moderate WBM in which the whole breast was hard but from the medial to the caudal regions of the breast was flexible. Score 3 represented a severe WBM in which the whole breast was extremely firm and rigid. 

### 2.3. Necropsy

A total of 144 birds (*n* = 6) for Trial 1 were necropsied from 56 to 59 d of age, the same was repeated in Trial 2. Two male birds per pen were selected for necropsy, including one normal bird without WBM and one bird affected by WBM. However, birds in some pens were either both normal or WBM-affected. The birds were weighed, euthanized by CO_2_ asphyxiation, and dissected. The internal organs (including liver, heart, gizzard, proventriculus, bursa of Fabricius, spleen, duodenum, jejunum, and ileum) were weighed. The lengths of the duodenum, jejunum, and ileum were measured. Gizzard and ileum pH were measured. The length of the footpad dermatitis was measured at the same time. Additionally, the length of the foot pad dermatitis of all broilers was measured and recorded at 47 d of age.

### 2.4. Processing

The Cobb 700 and Ross 708 broilers were, respectively, processed at ages 62 and 64 d. In each pen, 3 female and 3 male birds were selected, weighed, tagged, and withdrawn from feed 15 h before processing. The processing was conducted in the commercial processing plant at the Mississippi State University Poultry Research Farm following poultry processing standards [28]. Broilers were hung in plastic shackles and electrically stunned (11.5 volts, <0.5 mA AC to DC current for 3 s) at the speed of 22 broilers per minute. The birds were then manually cut immediately after stunning, and bleeding lasted for 140 s. Upon completion of exsanguination, the broilers were scalded at 53.3 °C for 191 s, picked for 35 s, and then mechanically eviscerated. The weights of carcasses (without neck, giblets, and abdominal fat pad) and abdominal fat pads (including leaf fat surrounding the cloaca and gizzard) were measured immediately after processing. The broiler carcasses were stored in ice water for 4 h. The carcasses were deboned into breasts (pectoralis major), tenderloins (pectoralis minor), wings, drumsticks, and thighs. Then, the weights of carcass parts were weighed and recorded. 

### 2.5. Gross Margin Return

Three methods were adopted for the calculation of gross marginal returns about cut-up carcasses. In one method, the calculation did not consider WBM incidence. In the other two methods, the percentage of severe WBM and the combined percentage of medium and severe WBM were considered in the calculation. The severe and medium WBM-affected meats would be at half the price of normal meat in the calculation. The method of calculating gross margin return was based on a previous study [21].

Gross marginal return = Price (the U.S. national average price) × Weights of the cut-up parts (breast, tenderloin, wing, drumstick, and thigh) – Feed price × Feed intake.

The prices of carcass cut-up parts were obtained from the website of the USDA on 6 December 2019 [29].

### 2.6. Statistical Analysis

A randomized complete block design with a factorial arrangement of 12 treatments (4 AA × 3 AME) was used in each trial. Dietary AA and AME were considered fixed effects. The block was considered a random effect in the model. The gender ratio in each pen was analyzed as a covariance factor. The internal organ weights, intestine length, WBM incidence, carcass parts weight of processing, and marginal returns were analyzed by using a two-way ANOVA of a PROC GLM procedure in SAS version 9.4. The normal and WBM broilers were separately analyzed in the sampling results. The normality of percentage data was evaluated by using the PROC UNIVARIATE procedure before analysis. Means that were significantly different at *p* < 0.05 were separated using the Tukey–Kramer comparison test. 

## 3. Results

### 3.1. Cobb 700 Broilers in Trial 1

#### 3.1.1. Internal Organ Development

Among normal birds without WBM, feeding broilers 70% and 80% AA reduced the gizzard pH compared to those fed 90% AA (*p* = 0.01, Table 5). The absolute weights of proventriculus, gizzard, pancreas, and heart were affected by the interaction of AA and AME, but the Tukey comparison test failed to separate treatment means.

Among birds affected by WBM, feeding broilers 84% AME decreased gizzard and duodenum weights compared with broilers fed 100% AME (*p* = 0.04, and 0.03, Table 6). 

Feeding broilers 70% AA with 100% AME reduced gizzard pH compared to those fed 90% AA with 92% AME and those fed 100% AA with 100% AME (*p* = 0.03, 0).

#### 3.1.2. Carcass Weight and Yield

Feeding broilers 70% AA increased broiler fat pad weight compared to those fed 90% and 100% AA (*p* < 0.01, Table 7); and the 70% AA-fed group had the lowest relative weight of carcass, breast, and tenderloin (all *p* < 0.01). Feeding broilers 70% and 80% AA increased the relative weight of the fat pad compared to that fed 90% and 100% AA (*p* < 0.01).

Feeding broilers 84% and 92% AME lowered the absolute and relative weight of the fat pad compared to those fed 100% AME (all *p* < 0.01). Feeding broilers 84% AME improved the relative weights of the carcass, breast, and tenderloin compared to those fed 100% AME (all *p* < 0.01).

Feeding broilers 70% AA with 100% AME lowered absolute breast weight compared to broilers of five treatments (*p* < 0.01); and lowered absolute wing weight as compared with two treatments (*p* = 0.02, Table 7). 

#### 3.1.3. Woody Breast Myopathy Incidence

Feeding broilers 70% AA increased the percentage of normal breasts compared to broilers fed 90% and 100% AA (*p* < 0.01) at d 33 and showed the highest percentage of normal breasts at 40, 47, and 62 days (all *p* < 0.01, Table 8). As compared with broilers fed 100% AA, 70% AA-fed group exhibited lower percentages of slight, medium, severe, and severe WBM, respectively, at 33, 40, 40, 47, and 62 d of age (all *p* < 0.01).

The 84% AME-fed groups showed a lower percentage of normal breast than 100% AME-fed birds at d 40 (*p* = 0.03) and d 47 (*p* = 0.04, Table 8). At d 62, 84% AME-fed birds exhibited the lowest percentage of normal breast (*p* < 0.01) and a higher percentage of slight WBM than 100% AME-fed broilers (*p* = 0.02).

#### 3.1.4. Footpad Dermatitis Length

As compared with 100% AA-fed broilers, the average footpad dermatitis length was decreased in 70% AA-fed Cobb 700 broilers at 47 d of age (*p* = 0.01) and normal broilers without WBM at 57 days (*p* = 0.04, Table 9). Feeding broilers 84% AME increased the average footpad dermatitis length as compared with broilers fed 92% AME at d 47 (*p* = 0.03).

#### 3.1.5. Economic Analysis

At 62 days, 70% AA-fed broilers decreased marginal return as compared with 90% and 100% AA-fed broilers (*p* = 0.01) when the incidence of WBM was not considered in the calculation (Table 9). In total, 84% AME-fed broilers increased the marginal return as compared with 100% AME-fed birds (*p* = 0.03).

### 3.2. Ross 708 Broilers in Trial 2

#### 3.2.1. Internal Organ Development

Among normal birds without WBM, 70% AA-fed broilers showed higher weights of proventriculus and bursa (*p* = 0.02 and 0.01) and a lower pH of the gizzard (*p* = 0.01) than those fed 100% AA (Table 10). The absolute weights and lengths of jejunum and ileum were affected by dietary AA reduction, but the Tukey comparison test failed to separate treatment means. For the spleen, 80% and 90% AA-fed broilers increased spleen as compared with 100% AA-fed broilers (*p* = 0.01); however, 92% AME-fed broilers decreased spleen as compared with 100% AME-fed broilers (*p* = 0.01). Broilers fed 84% AME had a significantly lower jejunum weight (*p* = 0.03) compared to those fed 100% AME. Among birds affected by WBM, feeding birds 84% AME decreased gizzard and jejunum absolute weights as compared with 100% AME-fed broilers (*p* = 0.05, and 0.03) (Table 11). 

#### 3.2.2. Carcass Weight and Relative Weight

Feeding broilers the AA-reduced diet (70%, 80%, and 90% AA) increased the fat pad absolute and relative weights as compared with those fed 100% AA birds (all *p* < 0.01, Table 12). Among all AA levels, broilers fed 70% AA showed the lowest absolute weights of carcass, wing, and breast among broilers fed all levels of AA (all *p* < 0.01). Broilers fed 70% AA exhibited the lowest wing and breast weights among broilers fed all levels of AA (all *p* < 0.01).

Broilers fed 84% AME exhibited a higher breast and a lower fat pad weight than those fed 100% AME (all *p* < 0.01). The 70% AA with 100% AME-fed broilers exhibited the lowest relative weight of carcass, breast, and tenderloin among all treatments (*p* < 0.01, <0.01, =0.02, Table 12).

#### 3.2.3. Woody Breast Incidence

The 70% and 80% AA-fed groups showed higher percentages of the normal breast WBM than 100% AA-fed broilers at d 40 (all *p* < 0.01) and d 47 (all *p* < 0.01). In addition, 70% AA-fed broilers showed the highest percentage of normal breast (*p* < 0.01) at d 64 (Table 13). Moreover, 70% and 80% AA-fed birds showed a lower percentage of moderate WBM than 100% AA-fed broilers at d 40 (all *p* < 0.01) and d 47 (all *p* < 0.01). 

The 84% AME-fed broilers showed a lower percentage of normal breast (*p* < 0.01) and a higher percentage of slight WBM (*p* < 0.01) than 100% AME-fed broilers at d 40. In addition, 84% of AME-fed broilers exhibited the lowest percentage of normal breast at d 40 (all *p* < 0.01) and d 64 (all *p* < 0.01); and the highest percentage of medium WBM and severe WBM, respectively, at d 40 and 64 (all *p* < 0.01).

At day 33, broilers fed 90% AA with 84% AME showed a lower proportion of normal breast compared to six other treatments (*p* < 0.01) and a higher proportion of slight WBM than seven other treatments (*p* = 0.01, Table 13), However, these differences were no longer significant after day 40.

#### 3.2.4. Footpad Dermatitis Length

Feeding Ross 708 broilers 70% and 80% AA decreased the average footpad dermatitis length as compared with that fed 100% AA at 47 d (*p* < 0.01) and normal broilers at 59 d (Table 14). Feeding broilers 84% AME increased the average footpad dermatitis length as compared with broilers fed 100% AME at d 47 (*p* < 0.01), normal broilers at day 59 (*p* < 0.01), and WBM-affected broilers at d 59 (*p* = 0.02). 

#### 3.2.5. Economic Analysis

At 64 d, 90% AA and 70% AA-fed Ross 708 broilers, respectively, exhibited the highest and the lowest marginal return among all broilers fed four levels of AA in three different methods of analysis (all *p* < 0.01, Table 14). Ross 708 broilers fed 84% AME achieved the highest marginal return across all three AME levels under three different conditions (all *p* < 0.01). 

## 4. Discussion

### 4.1. Internal Organ Development

The proventriculus development was promoted by dietary AA reduction. The 30% reduction in AA in diets leads to substantial development of the proventriculus. The increased weight of proventriculus may be largely due to the broilers’ increased feed intake as compensation for dietary AA dilution. The proventriculus was stimulated to produce more H^+^ (hydrogen iron) by the increased feed intake. The H^+^ mixed with feed and moved forward into the gizzard. As expected, the increased production of H^+^ was confirmed by a lower gizzard pH in both strain broilers fed 70% AA in the current study. The low gizzard pH is required to activate the proteinase (pepsin) and hydrolyze protein. Decreased pH may improve nutrient utilization and feed efficiency. In a companion study, the FCR of Ross 708 broilers fed a 30% AA reduced diet was improved from 48 to 64 days [30]. Therefore, broilers could stimulate proventriculus function to improve H^+^ production in response to dietary AA reduction.

Intestinal development responds differently to dietary AA and AME reduction. Among normal broilers, dietary 30% AA reduction increased the jejunum weight of Ross 708 broilers at 58 days. However, the normal Ross 708 broilers fed 84% AME exhibited lower jejunum weight than the broilers fed 100% AME at 58 days. The increased jejunum weight result from dietary AA reduction is different from previous week-old chicken studies. As dietary essential AAs were reduced, intestinal development was inhibited in 6–9 day broilers [31] and 7-day broilers [32] since essential AAs are involved in mucosal protein synthesis for the intestine. One possible explanation was that old broilers were less sensitive to dietary AA reduction since their AA requirements decreased with age [24,25]. In the current study, the improvement in intestinal development could be that the broilers have adapted to the AA reduction since they were continuously fed an AA-reduced diet from 0 to 58 days. In contrast, the energy requirement increases with the age of broilers [24,25]. This may explain the slow development of the intestinal segments due to the dietary reduction in AME. These results suggest that aged broiler birds are more sensitive to changes in dietary AME compared to AA.

### 4.2. Carcass Weights and Yields

Dietary 30% AA reduction decreased breast, tenderloin, and carcass relative weights of Cobb 700 broilers; and decreased carcass, wing, and breast absolute weights of Ross 708 birds. Previous studies have investigated the effects of dietary AA levels on broiler processing and reported similar conclusions [21,33]. However, responses of the breast, tenderloin, and carcass weights to AA reduction were different from that of thigh weight in our study: AA reduction did not affect the related thigh weight. Thigh muscles are used in daily exercise to protect themselves from atrophy caused by nutritional restriction, whereas breast muscles in broilers are the product of genetic selection [34] and have little functional purpose. Breasts can store large amounts of protein and can decompose themselves to provide amino acids when nutrients are limited. Breast meats and whole-leg (thigh and drumstick) meats differ in their nutrient compositions, including crude protein and AAs [35,36]. Breast and leg meats have different nutrient requirements and respond differently to the same dietary AA reduction [37]. The different responses between breast and thigh meat are also due to different meat types (fast-twitch or slow-twitch). Breast meat is the type IIB of fast-twitch, whereas thigh and drumstick meats are the type I and IIA of slow-twitch. Studies in rats have shown [38] that fast-twitch muscles respond the most to nutritional restrictions, while slow-twitch muscles are insensitive to that. Tesseraud, et al. [39] also reported this phenomenon in broiler breast muscle.

Dietary 30% AA reduction increased the fat pad absolute weight of two strains. These results were consistent with a previous study that the abdominal fat pad of broilers fed a TSAA-deficient diet was increased by 28% compared with that of broilers fed a control diet [11]. Variations in the metabolism of AAs due to insufficient essential AAs may be responsible for increased fat pad weight. In the current study, the first three limiting AAs (TSAA, Lys, and Thr) were decreased by 30%. However, a deficiency of these AAs would decrease protein synthesis in broilers [40,41]. As a result, other sufficient AAs could not be used for protein synthesis and may be converted into fat through transamination or deamination reactions. Additionally, fat deposition requires more energy than protein deposition [42]; thereby, the increased fat pad weight might reflect decreased feed utilization. In this work companion study, Ross 708 broilers fed 70% AA exhibited a higher FCR than broilers fed 90% and 100% AA from 0 to 41 days [30].

Previous research has shown that reducing dietary energy as a nutrient in broiler diets causes a decrease in fat pad weight [9]. In the present study, both Cobb 700 and Ross 708 broilers fed 84% AME decreased fat pad weight compared with broilers fed 100% AME at days 62 and 64. The decreased fat pad weight in response to dietary energy reduction is largely due to reduced energy intake [43]. The fat pad deposition is decreased by dietary energy reduction [8]. 

### 4.3. Woody Breast Myopathy Incidence

Dietary AA reduction has been suggested to control WBM incidence in a previous study [3]. In the current study, both two strains of broilers fed 70% AA exhibited lower percentages of slight WBM and moderate WBM than the broilers fed 100% AA at 40 and 47 days. The WBM incidence was decreased at d 42 when dietary lysine was reduced by 30% during either from 12 to 28 d or 28 to 42 days [3]. We also found that when dietary AA was reduced by 30%, the reduction in WBM incidence was accompanied by the reduction in BW, which is consistent with the findings of Cruz, et al. [3] and Zhang, et al. [26]. AA-restricted diets did not maximize the genetic potential of broilers and reduced BW, but they did allow normal tissue renewal to be supported. 

However, dietary AME reduction was linked with an increased percentage of WBM incidence. Ross 708 broilers increased moderate WBM percentage at 47 days when birds were fed a 16% AME-reduced diet. Conversely, a previous study reported that broilers fed a low-energy diet (3140 and 3175 kcal/kg from 29 to 36 d and from 37 to 46 d, respectively) had a lower percentage of mild WBM than broilers fed a medium-energy diet (3175 and 3210 kcal/kg, respectively) [44]. The discrepancy between our results and those from Maynard, et al. [44] is largely due to the different responses to breast weight. Although dietary AME reduction was more severe in the current study (16%) than in the previous study (1.1%), Maynard et al. (2019) [43] observed that breast weight was not affected by dietary AME reduction, whereas the current study found that broilers fed 84% AME increased breast relative weight. The increased breast relative weight may have worsened WBM conditions in broilers considering that broilers with heavy breasts were more likely to develop WBM [3].

### 4.4. Footpad Dermatitis Length

In the current study, both Cobb 700 and Ross 708 broilers fed 70% AA exhibited a shorter footpad dermatitis length compared with the broilers fed 90% and 100% AA at 47 days. A previous study also found that the footpad dermatitis severity was decreased in broilers fed a 3% protein-reduced diet in which the concentrations of Lys, TSAA, and Thr were the same as a control treatment [10]. The reduced footpad dermatitis is likely associated with decreased dietary crude protein and soybean meal in the AA-reduced diet. Soybean meal is the most common protein source in the diets of broilers. However, soybean meal has been found to contain a sticky indigestible non-starch polysaccharide that might cause footpad dermatitis [45,46]. In the current study, the AA-reduced diet had a low concentration of soybean meal and crude protein. Therefore, the footpad dermatitis was improved by dietary 30% AA reduction.

In contrast, dietary AME reduction did not reduce the incidence of footpad dermatitis. Both strains of broilers fed 84% AME increased the length of footpad dermatitis compared with broilers fed 92% AME. However, Bilgili, et al. [47] reported that the footpad dermatitis of broilers fed a low-energy diet (3086 kcal/kg) was not affected as compared with broilers fed a high-energy diet (3187 kcal/kg). The inconsistency between these results may be due to different litter conditions, which would influence the development of footpad dermatitis [46]. Bilgili, et al. [47] used new litter in the research, and no footpad dermatitis was detected at d 49. These results may have suggested that litter and feed management could control footpad dermatitis. 

### 4.5. Economic Return

The marginal return must find an equilibrium between the cost of feed and the yield of the carcass, both of which might be diminished due to a reduction in dietary nutrients. For Ross 708 broilers, marginal return increased when dietary AA decreased from 100% to 90%, but then marginal return decreased rapidly as AA decreased to 70%. Cobb 700 broilers fed 70% AA showed decreased marginal returns. Additionally, both strains of broilers fed 84% AME had higher marginal returns compared with the broilers fed 100% AME. 

The broiler will adjust the feed intake according to the energy content of the feed to keep the total energy intake relatively constant to meet the demand, but the relative total protein intake of the broiler will be affected by the protein–energy ratio (PER) of the feed [43]. In this study, reducing the content of feed AA will lead to a decrease in the protein–energy ratio, and reducing the content of feed AME will increase the protein–energy ratio. The decrease in feed protein–energy ratio will reduce the broiler BW, carcass, and breast weight, while the weight of the fat pad will increase. Insufficient dietary protein will hinder muscle synthesis, and more energy will be deposited in the form of fat pads, thereby reducing marginal return. On the contrary, increasing the feed protein–energy ratio, although not directly increasing the broiler BW, will reduce the fat pad weight and increase the absolute and relative weight of carcass, breast, and tenderloin. Feed is converted into higher-value chicken products, thereby improving marginal return. The WBM-affected meat decreased meat quality and would be sold at a lower price. Increasing the relative AA content in the feed would increase the WBM incidence of broilers, leading to reduced breast meat quality and diminishing marginal returns, but given the relatively low overall incidence of WBM, the increase in AA levels can still enhance carcass yield, thereby improving the final marginal economic benefit.

The AA and AME interaction results show that 90%AA-84%AME-fed Ross 708 and Cobb 700 broilers have the best marginal return, and the feed protein–energy ratio was both increased in these two treatments. This finding suggests that lowering the relative level of dietary AME, thereby increasing the relative AA content, can enhance the final economic return by improving the yield of high-value chicken cut-up parts. Compared to diets with relatively higher AME, which tend to promote excess energy deposition as fat, this approach offers greater value to the broiler industry through improved carcass composition and profitability. Therefore, we recommend feeding the following: AME: 12.70 MJ/kg and CP: 25.12% (PER: 19.78 g/MJ) for Starter (0–10 d); AME: 13.00 MJ/kg and CP: 22.77% (PER: 17.51 g/MJ) for Grower (11–24 d); AME: 13.39 MJ/kg and CP: 21.46% (PER: 16.03 g/MJ) for Finisher (25–39 d); AME: 13.49 MJ/kg and CP: 20.57% (PER: 15.25 g/MJ) for Withdrawal (40–63 d) to improve economic returns for broiler companies. 

## 5. Conclusions

AA reductions by 30% could control WBM incidence and footpad dermatitis but result in decreased carcass quality and economic returns. Reducing AME to 84%, although increasing the WBM incidence, could reduce fat deposition and improve meat production and marginal return. We recommend a feed protein–energy ratio of 19.78 g/MJ for Starter (0–10 d); 17.51 g/MJ for Grower (11–24 d); 16.03 g/MJ for Finisher (25–39 d); 15.25 g/MJ for Withdrawal (40–63 d) to improve economic returns.

## Figures and Tables

**Table 1 animals-15-01064-t001:** Feed ingredient and nutrient composition of 12 dietary treatments with factorial combinations of 4 levels of digestible amino acids and 3 levels of apparent metabolizable energy during the Starter (d 0–10) feeding phase.

Parameter	Treatment											
AA (%) ^1^	70	70	70	80	80	80	90	90	90	100	100	100
AME (%)	84	92	100	84	92	100	84	92	100	84	92	100
Yellow Corn %	56.96	67.91	71.06	57.32	64.14	63.94	52.05	60.38	56.74	48.28	55.10	49.55
Soybean Meal %	29.90	22.74	22.27	29.38	28.24	28.27	35.14	33.74	34.35	40.64	39.50	40.43
Soybean Oil %	0.00	0.50	2.21	0.00	0.50	3.35	0.50	0.50	4.49	0.50	1.01	5.64
DL-Methionine %	0.18	0.23	0.23	0.29	0.28	0.28	0.33	0.32	0.33	0.38	0.37	0.38
L-Lysine HCl %	0.03	0.23	0.24	0.21	0.23	0.23	0.20	0.22	0.21	0.19	0.21	0.20
L-Threonine %	0.00	0.09	0.09	0.09	0.10	0.10	0.10	0.10	0.10	0.11	0.11	0.11
Ronozyme %	0.02	0.02	0.02	0.02	0.02	0.02	0.02	0.02	0.02	0.02	0.02	0.02
Dicalcium Phosphate %	1.74	1.75	1.74	1.74	1.73	1.73	1.72	1.71	1.72	1.70	1.69	1.70
Limestone %	1.45	1.47	1.48	1.44	1.45	1.45	1.42	1.44	1.43	1.41	1.42	1.41
Salt %	0.40	0.33	0.33	0.33	0.33	0.33	0.33	0.32	0.33	0.33	0.32	0.33
Premix % ^2^	0.25	0.25	0.25	0.25	0.25	0.25	0.25	0.25	0.25	0.25	0.25	0.25
Choline Chloride %	0.06	0.09	0.09	0.06	0.06	0.06	0.03	0.03	0.03	0.00	0.00	0.00
Sand %	9.00	4.39	0.00	8.87	2.68	0.00	7.90	0.96	0.00	6.19	0.00	0.00
Feed Price (USD/kg)	0.24	0.24	0.26	0.25	0.26	0.28	0.26	0.27	0.29	0.27	0.28	0.31
Calculated Composition												
CP, %	16.48	16.46	16.49	18.77	18.79	18.79	21.10	21.11	21.11	23.42	23.44	23.43
Ca, %	0.96	0.96	0.96	0.96	0.96	0.96	0.96	0.96	0.96	0.96	0.96	0.96
Available *p* %	0.48	0.48	0.48	0.48	0.48	0.48	0.48	0.48	0.48	0.48	0.48	0.48
M.E. (kcal/kg)	2550	2792	3035	2549	2792	3035	2549	2792	3035	2549	2792	3035
Digestible Met %	0.44	0.46	0.46	0.54	0.53	0.53	0.61	0.60	0.61	0.68	0.68	0.68
Digestible TSAAs %	0.66	0.66	0.66	0.76	0.76	0.76	0.85	0.85	0.85	0.95	0.95	0.95
Digestible Lys %	0.90	0.90	0.90	1.02	1.02	1.02	1.15	1.15	1.15	1.28	1.28	1.28
Digestible Thr %	0.60	0.60	0.60	0.69	0.69	0.69	0.77	0.77	0.77	0.86	0.86	0.86
Digestible Try %	0.22	0.18	0.18	0.21	0.21	0.21	0.24	0.24	0.24	0.27	0.27	0.27
Digestible Leu %	1.47	1.33	1.34	1.45	1.48	1.47	1.60	1.62	1.61	1.74	1.77	1.75
Digestible Val %	0.78	0.67	0.67	0.77	0.77	0.77	0.86	0.86	0.86	0.96	0.96	0.96
Digestible Arg %	1.15	0.96	0.96	1.14	1.13	1.13	1.30	1.29	1.29	1.47	1.45	1.46
Choline (ppm)	789	789	789	789	789	789	789	789	789	789	788	789
Chloride %	0.27	0.27	0.27	0.26	0.26	0.26	0.25	0.25	0.25	0.24	0.24	0.24
Sodium %	0.19	0.16	0.16	0.16	0.16	0.16	0.16	0.16	0.16	0.16	0.16	0.16
ME/CP (kcal/kg/%)	136.3	169.6	184.1	135.8	148.6	161.5	120.8	132.3	143.8	108.9	119.1	129.5

^1^ Amino acids in the 100% diet were at the higher recommended levels of digestible amino acids (lysine, TSAAs, and threonine). ^2^ Premix did not contain riboflavin and provided the following per kilogram of finished diet: retinyl acetate, 2.654 μg; cholecalciferol, 110 μg; DL-α-tocopherol acetate, 9.9 mg; menadione, 0.9 mg; vitamin B12, 0.01 mg; folic acid, 0.6 μg; choline, 379 mg; D-pantothenic acid, 8.8 mg; niacin, 33 mg; thiamine, 1.0 mg; D-biotin, 0.1 mg; pyridoxine, 0.9 mg; ethoxyquin, 28 mg; manganese, 55 mg; zinc, 50 mg; iron, 28 mg; copper, 4 mg; iodine, 0.5 mg; selenium, 0.1 mg.

**Table 2 animals-15-01064-t002:** Feed ingredient and nutrient composition of 12 dietary treatments with factorial combinations of 4 levels of digestible amino acids and 3 levels of apparent metabolizable energy during the Grower (d10–24) feeding phase.

Parameter	Treatment											
AA (%) ^1^	70	70	70	80	80	80	90	90	90	100	100	100
AME (%)	84	92	100	84	92	100	84	92	100	84	92	100
Yellow Corn %	62.18	72.39	74.46	62.24	69.27	68.45	57.33	65.86	61.93	53.92	61.10	55.41
Soybean Meal %	25.00	19.17	18.82	24.90	23.73	23.86	30.14	28.71	29.37	35.13	33.93	34.88
Soybean Oil %	0.00	0.50	2.65	0.00	0.50	3.62	0.50	0.50	4.66	0.50	0.94	5.69
DL-Methionine %	0.17	0.20	0.20	0.26	0.25	0.25	0.30	0.29	0.30	0.34	0.34	0.34
L-Lysine HCl %	0.06	0.22	0.23	0.21	0.23	0.22	0.20	0.22	0.21	0.19	0.21	0.19
L-Threonine %	0.00	0.07	0.07	0.08	0.08	0.08	0.09	0.09	0.09	0.09	0.09	0.09
Ronozyme %	0.02	0.02	0.02	0.02	0.02	0.02	0.02	0.02	0.02	0.02	0.02	0.02
Dicalcium Phosphate %	1.51	1.51	1.51	1.51	1.50	1.50	1.49	1.48	1.48	1.48	1.46	1.47
Limestone %	1.36	1.38	1.38	1.35	1.36	1.36	1.33	1.35	1.34	1.32	1.33	1.32
Salt %	0.40	0.33	0.33	0.33	0.33	0.33	0.33	0.33	0.33	0.33	0.32	0.33
Premix % ^2^	0.25	0.25	0.25	0.25	0.25	0.25	0.25	0.25	0.25	0.25	0.25	0.25
Choline Chloride %	0.05	0.08	0.08	0.06	0.06	0.06	0.03	0.03	0.03	0.00	0.00	0.00
Sand %	9.00	3.88	0.00	8.80	2.43	0.00	7.99	0.88	0.00	6.44	0.00	0.00
Feed Price (USD/kg)	0.23	0.24	0.25	0.24	0.25	0.27	0.25	0.26	0.28	0.26	0.27	0.30
Calculated Composition												
CP %	15.09	15.09	15.09	17.02	17.03	17.03	19.13	19.14	19.13	21.23	21.25	21.24
Ca %	0.87	0.87	0.87	0.87	0.87	0.87	0.87	0.87	0.87	0.87	0.87	0.87
Available *p* %	0.44	0.44	0.43	0.43	0.43	0.44	0.44	0.44	0.44	0.43	0.43	0.43
M.E. (kcal/kg)	2611	2859	3108	2611	2859	3108	2611	2859	3108	2611	2859	3108
Digestible Met %	0.40	0.42	0.42	0.49	0.48	0.49	0.55	0.55	0.55	0.62	0.62	0.62
Digestible TSAA %	0.61	0.61	0.61	0.70	0.70	0.70	0.78	0.78	0.78	0.87	0.87	0.87
Digestible Lys %	0.80	0.80	0.81	0.92	0.92	0.92	1.03	1.04	1.03	1.15	1.15	1.15
Digestible Thr %	0.54	0.54	0.54	0.62	0.62	0.62	0.69	0.69	0.69	0.77	0.77	0.77
Digestible Try %	0.19	0.16	0.16	0.19	0.19	0.19	0.22	0.22	0.22	0.25	0.24	0.24
Digestible Leu %	1.35	1.25	1.26	1.35	1.37	1.37	1.48	1.51	1.49	1.61	1.63	1.62
Digestible Val %	0.70	0.62	0.62	0.70	0.70	0.70	0.78	0.78	0.78	0.87	0.87	0.87
Digestible Arg %	1.02	0.86	0.86	1.01	1.00	1.00	1.16	1.15	1.15	1.31	1.31	1.31
Choline (ppm)	729	729	729	729	729	729	729	729	729	730	729	730
Chloride %	0.27	0.27	0.27	0.26	0.26	0.26	0.25	0.25	0.25	0.24	0.24	0.24
Sodium %	0.19	0.16	0.16	0.16	0.16	0.16	0.16	0.16	0.16	0.16	0.16	0.16
ME/CP (kcal/kg/%)	155.3	189.5	206.0	153.4	167.9	182.5	136.5	149.4	162.5	123.0	134.5	146.3

^1^ Amino acids in the 100% diet were at the higher recommended levels of digestible amino acids (lysine, TSAAs, and threonine). ^2^ Premix did not contain riboflavin and provided the following per kilogram of finished diet: retinyl acetate, 2.654 μg; cholecalciferol, 110 μg; DL-α-tocopherol acetate, 9.9 mg; menadione, 0.9 mg; vitamin B12, 0.01 mg; folic acid, 0.6 μg; choline, 379 mg; D-pantothenic acid, 8.8 mg; niacin, 33 mg; thiamine, 1.0 mg; D-biotin, 0.1 mg; pyridoxine, 0.9 mg; ethoxyquin, 28 mg; manganese, 55 mg; zinc, 50 mg; iron, 28 mg; copper, 4 mg; iodine, 0.5 mg; selenium, 0.1 mg.

**Table 3 animals-15-01064-t003:** Feed ingredient and nutrient composition of 12 dietary treatments with factorial combinations of 4 levels of digestible amino acids and 3 levels of apparent metabolizable energy during the Finisher (d 24–41) feeding phase.

Parameter	Treatment											
AA (%) ^1^	70	70	70	80	80	80	90	90	90	100	100	100
AME (%)	84	92	100	84	92	100	84	92	100	84	92	100
Yellow Corn %	67.99	75.30	73.54	64.37	72.78	68.94	61.57	69.16	63.72	58.76	63.94	58.50
Soybean Meal %	19.96	18.67	18.98	23.28	22.36	22.78	27.38	26.55	27.15	31.48	30.92	31.51
Soybean Oil %	0.00	0.50	4.01	0.50	0.50	4.77	0.50	0.78	5.61	0.50	1.63	6.45
DL-Methionine %	0.16	0.15	0.15	0.21	0.20	0.20	0.25	0.24	0.25	0.30	0.29	0.30
L-Lysine HCl %	0.09	0.11	0.11	0.12	0.13	0.13	0.13	0.13	0.13	0.13	0.14	0.13
L-Threonine %	0.00	0.01	0.01	0.03	0.02	0.03	0.04	0.03	0.04	0.05	0.05	0.05
Ronozyme %	0.02	0.02	0.02	0.02	0.02	0.02	0.02	0.02	0.02	0.02	0.02	0.02
Dicalcium Phosphate %	1.28	1.26	1.27	1.27	1.25	1.26	1.25	1.24	1.25	1.24	1.23	1.24
Limestone %	1.26	1.27	1.27	1.25	1.26	1.25	1.23	1.25	1.24	1.22	1.23	1.22
Salt %	0.33	0.33	0.33	0.33	0.33	0.33	0.33	0.33	0.33	0.33	0.33	0.33
Premix % ^2^	0.25	0.25	0.25	0.25	0.25	0.25	0.25	0.25	0.25	0.25	0.25	0.25
Choline Chloride %	0.06	0.06	0.06	0.05	0.04	0.04	0.02	0.02	0.02	0.00	0.00	0.00
Sand %	8.60	2.06	0.00	8.32	0.86	0.00	7.02	0.00	0.00	5.72	0.00	0.00
Feed Price (USD/kg)	0.22	0.23	0.25	0.23	0.24	0.26	0.24	0.25	0.28	0.25	0.26	0.29
Calculated Composition												
CP %	14.92	14.90	14.91	16.28	16.50	16.40	18.03	18.23	18.09	19.78	19.92	19.77
Ca %	0.78	0.78	0.78	0.78	0.78	0.78	0.78	0.78	0.78	0.78	0.78	0.78
Available *p* %	0.39	0.39	0.39	0.39	0.39	0.39	0.39	0.39	0.39	0.39	0.39	0.39
M.E. (kcal/kg)	2689	2944	3200	2688	2944	3200	2688	2944	3200	2688	2944	3200
Digestible Met %	0.37	0.37	0.37	0.44	0.43	0.43	0.50	0.49	0.50	0.56	0.56	0.56
Digestible TSAA %	0.56	0.56	0.56	0.64	0.64	0.64	0.72	0.72	0.72	0.80	0.80	0.80
Digestible Lys %	0.71	0.71	0.71	0.82	0.82	0.82	0.92	0.92	0.92	1.02	1.02	1.02
Digestible Thr %	0.48	0.48	0.48	0.54	0.54	0.54	0.61	0.61	0.61	0.68	0.68	0.68
Digestible Try %	0.17	0.16	0.16	0.18	0.18	0.18	0.20	0.21	0.20	0.23	0.23	0.23
Digestible Leu %	1.24	1.26	1.25	1.31	1.36	1.34	1.42	1.46	1.44	1.53	1.56	1.53
Digestible Val %	0.62	0.62	0.62	0.67	0.68	0.68	0.74	0.75	0.75	0.81	0.82	0.81
Digestible Arg %	0.87	0.86	0.86	0.97	0.97	0.97	1.09	1.09	1.09	1.21	1.21	1.21
Choline (ppm)	693	693	693	693	693	693	693	693	693	693	696	693
Chloride %	0.24	0.24	0.24	0.24	0.24	0.24	0.23	0.23	0.23	0.23	0.23	0.23
Sodium %	0.16	0.16	0.16	0.16	0.16	0.16	0.16	0.16	0.16	0.16	0.16	0.16
ME/CP (kcal/kg/%)	180.2	197.6	214.6	165.1	178.4	195.1	149.1	161.5	176.9	135.9	147.8	161.9

^1^ Amino acids in the 100% diet were at the higher recommended levels of digestible amino acids (lysine, TSAAs, and threonine). ^2^ Premix did not contain riboflavin and provided the following per kilogram of finished diet: retinyl acetate, 2.654 μg; cholecalciferol, 110 μg; DL-α-tocopherol acetate, 9.9 mg; menadione, 0.9 mg; vitamin B12, 0.01 mg; folic acid, 0.6 μg; choline, 379 mg; D-pantothenic acid, 8.8 mg; niacin, 33 mg; thiamine, 1.0 mg; D-biotin, 0.1 mg; pyridoxine, 0.9 mg; ethoxyquin, 28 mg; manganese, 55 mg; zinc, 50 mg; iron, 28 mg; copper, 4 mg; iodine, 0.5 mg; selenium, 0.1 mg.

**Table 4 animals-15-01064-t004:** Feed ingredient and nutrient composition of 12 dietary treatments with factorial combinations of 4 levels of digestible amino acids and 3 levels of apparent metabolizable energy during the Withdrawal (d 41–64) feeding phase.

Parameter	Treatment											
AA (%) ^1^	70	70	70	80	80	80	90	90	90	100	100	100
AME (%)	84	92	100	84	92	100	84	92	100	84	92	100
Yellow Corn	68.69	77.60	73.34	67.61	76.08	70.69	64.94	71.21	65.74	62.28	66.25	60.78
Soybean Meal	19.84	18.27	19.02	21.42	20.49	21.08	25.32	24.63	25.23	29.22	28.78	29.38
Soybean Oil	0.00	0.00	4.37	0.00	0.00	4.83	0.00	0.77	5.63	0.00	1.57	6.43
DL-Methionine	0.12	0.11	0.12	0.18	0.17	0.18	0.22	0.21	0.22	0.26	0.26	0.27
L-Lysine HCl	0.04	0.07	0.05	0.11	0.12	0.12	0.12	0.12	0.12	0.12	0.12	0.12
L-Threonine	0.00	0.00	0.00	0.02	0.01	0.02	0.03	0.02	0.03	0.04	0.03	0.04
Ronozyme	0.02	0.02	0.02	0.02	0.02	0.02	0.02	0.02	0.02	0.02	0.02	0.02
Dicalcium Phosphate	1.22	1.21	1.21	1.21	1.20	1.21	1.20	1.19	1.20	1.19	1.18	1.19
Limestone	1.24	1.25	1.24	1.23	1.24	1.24	1.22	1.23	1.22	1.21	1.21	1.20
Salt	0.33	0.33	0.33	0.33	0.33	0.33	0.33	0.33	0.33	0.33	0.33	0.33
Premix ^2^	0.25	0.25	0.25	0.25	0.25	0.25	0.25	0.25	0.25	0.25	0.25	0.25
Choline Chloride	0.05	0.05	0.05	0.04	0.04	0.04	0.02	0.02	0.02	0.00	0.00	0.00
Sand	8.20	0.84	0.000.00	7.57	0.05	0.00.	6.33	0.00	0.00	5.10	0.00	0.00
Feed Price (USD/kg)	0.22	0.23	0.25	0.23	0.23	0.26	0.23	0.25	0.27	0.24	0.26	0.29
Calculated Composition												
CP, %	14.84	14.83	14.83	15.62	15.85	15.70	17.28	17.45	17.31	18.95	19.05	18.91
Ca, %	0.76	0.76	0.76	0.76	0.76	0.76	0.76	0.76	0.76	0.76	0.76	0.76
Available *p*, %	0.38	0.38	0.38	0.38	0.38	0.38	0.38	0.38	0.38	0.38	0.38	0.38
M.E. (kcal/kg)	2709	2967	3225	2709	2967	3225	2709	2967	3225	2709	2967	3225
Digestible Met, %	0.33	0.33	0.33	0.40	0.40	0.40	0.46	0.46	0.46	0.52	0.52	0.52
Digestible TSAA, %	0.53	0.53	0.53	0.60	0.60	0.60	0.67	0.67	0.67	0.75	0.75	0.75
Digestible Lys, %	0.67	0.67	0.67	0.77	0.77	0.77	0.86	0.86	0.86	0.96	0.96	0.96
Digestible Thr, %	0.47	0.47	0.47	0.51	0.51	0.51	0.58	0.58	0.58	0.64	0.64	0.64
Digestible Try	0.17	0.16	0.16	0.17	0.17	0.17	0.19	0.20	0.19	0.22	0.22	0.22
Digestible Leu	1.24	1.26	1.25	1.28	1.32	1.30	1.39	1.42	1.39	1.49	1.51	1.48
Digestible Val	0.62	0.62	0.62	0.65	0.65	0.65	0.71	0.72	0.71	0.78	0.79	0.78
Digestible Arg	0.87	0.85	0.86	0.92	0.92	0.92	1.04	1.03	1.03	1.15	1.15	1.15
Choline (ppm)	670	670	670	670	670	670	670	670	670	671	674	671
Chloride, %	0.22	0.23	0.23	0.24	0.23	0.24	0.23	0.23	0.23	0.22	0.22	0.22
Sodium, %	0.16	0.16	0.16	0.16	0.16	0.16	0.16	0.16	0.16	0.16	0.16	0.16
ME/CP (kcal/kg/%)	182.5	200.1	217.5	173.4	187.2	205.4	156.8	170.0	186.3	143.0	155.7	170.5

^1^ Amino acids in the 100% diet were at the higher recommended levels of digestible amino acids (lysine, TSAAs, and threonine). ^2^ Premix did not contain riboflavin and provided the following per kilogram of finished diet: retinyl acetate, 2.654 μg; cholecalciferol, 110 μg; DL-α-tocopherol acetate, 9.9 mg; menadione, 0.9 mg; vitamin B12, 0.01 mg; folic acid, 0.6 μg; choline, 379 mg; D-pantothenic acid, 8.8 mg; niacin, 33 mg; thiamine, 1.0 mg; D-biotin, 0.1 mg; pyridoxine, 0.9 mg; ethoxyquin, 28 mg; manganese, 55 mg; zinc, 50 mg; iron, 28 mg; copper, 4 mg; iodine, 0.5 mg; selenium, 0.1 mg.

**Table 5 animals-15-01064-t005:** Internal organ weights, intestine length, and pH of Cobb 700 broilers without woody breast at 56 d of age.

Treatment		Weight (g)											Length (cm)			pH	
AA (%)	AME (%)	BW	Prov ^1,2^	Gizzard ^2^	Pancreas ^2^	Heart ^2^	Liver	Spleen	Bursa	Duo ^1^	Jej ^1^	Ileum	Duo ^1,2^	Jej ^1^	Ileum	Gizzard	Ileum
70		3512	12.3	39.7	4.65	16.0	71.4	2.81	2.96	15.1	26.8	20.3	30.5	72.5	75.1	2.69 ^b^	6.51
80		3684	11.4	38.1	4.66	16.6	65.9	2.59	3.43	14.3	24.7	18.0	31.6	72.9	71.8	2.77 ^b^	6.57
90		3814	11.3	39.5	4.90	16.3	66.8	2.81	3.01	14.5	24.4	18.5	31.4	73.2	75.3	3.18 ^a^	6.31
100		3875	11.3	40.6	5.00	14.9	69.8	2.68	2.77	14.4	25.1	19.5	30.5	72.4	74.8	3.07 ^ab^	6.47
SEM ^3^		95.3	0.56	1.49	0.22	0.77	3.09	0.24	0.30	0.65	1.18	1.07	0.69	2.15	2.49	0.11	0.15
	84	3754	11.8	39.4	5.09	15.9	72.1	2.72	2.54	14.6	26.2	19.0	30.9	73.5	72.0	2.97	6.46
	92	3606	11.0	38.4	4.70	15.2	65.5	2.61	3.40	13.6	24.0	18.6	30.4	69.9	72.2	2.87	6.60
	100	3804	12.0	40.7	4.61	16.7	67.8	2.84	3.19	15.6	25.6	19.6	31.6	74.8	78.6	2.94	6.34
	SEM	82.5	0.48	1.29	0.19	0.67	2.68	0.20	0.26	0.56	1.02	0.93	0.59	1.86	2.16	0.09	0.13
70	84	3805 ^abc^	11.7 ^a^	42.6 ^a^	5.24 ^a^	16.9 ^a^			3.42 ^ab^				30.3 ^a^				
70	92	3469 ^abc^	12.6 ^a^	37.0 ^a^	4.19 ^a^	14.8 ^a^			2.64 ^ab^				29.3 ^a^				
70	100	3263 ^c^	12.6 ^a^	39.5 ^a^	4.52 ^a^	16.2 ^a^			2.83 ^ab^				31.8 ^a^				
80	84	3655 ^abc^	11.8 ^a^	34.8 ^a^	4.60 ^a^	16.8 ^a^			2.18 ^b^				29.4 ^a^				
80	92	3652 ^abc^	11.2 ^a^	40.4 ^a^	5.07 ^a^	17.1 ^a^			4.75 ^a^				33.0 ^a^				
80	100	3746 ^abc^	11.3 ^a^	39.1 ^a^	4.32 ^a^	16.0 ^a^			3.37 ^ab^				32.3 ^a^				
90	84	3396 ^bc^	10.0 ^a^	35.2 ^a^	4.65 ^a^	13.4 ^a^			1.63 ^b^				30.8 ^a^				
90	92	3852 ^abc^	10.2 ^a^	41.7 ^a^	5.06 ^a^	16.2 ^a^			3.54 ^ab^				30.3 ^a^				
90	100	4194 ^a^	13.8 ^a^	41.7 ^a^	5.00 ^a^	19.3 ^a^			3.86 ^ab^				33.2 ^a^				
100	84	4159 ^ab^	13.7 ^a^	45.0 ^a^	5.88 ^a^	16.6 ^a^			2.95 ^ab^				33.3 ^a^				
100	92	3452 ^abc^	9.9 ^a^	34.3 ^a^	4.50 ^a^	12.9 ^a^			2.66 ^ab^				28.9 ^a^				
100	100	4014 ^abc^	10.5 ^a^	42.5 ^a^	4.63 ^a^	15.2 ^a^			2.69 ^ab^				29.3 ^a^				
SEM		165	0.97	2.58	0.38	1.34			0.53				1.19				
*p*-value																	
AA		0.07	0.50	0.68	0.68	0.43	0.45	0.90	0.47	0.76	0.48	0.40	0.60	0.99	0.76	0.01	0.69
AME		0.60	0.50	0.58	0.35	0.44	0.71	0.83	0.07	0.11	0.41	0.79	0.47	0.55	0.05	1.00	0.48
AA × AME		<0.01	0.01	0.01	0.03	0.05	0.07	0.09	0.01	0.65	0.55	0.17	0.01	0.51	0.78	0.76	0.08

^a–c^ Means in a column not sharing a common superscript were different (*p* < 0.05). ^1^ Prov stands for proventriculus, duo stands for duodenum, and Jej stands for jejunum. ^2^ Tukey’s test was not able to separate treatment means of these measured variables. ^3^ SEM = standard error of mean.

**Table 6 animals-15-01064-t006:** Internal organ weights, intestine length, and pH of Cobb 700 broilers with WBM at 57 d of age.

Treatment		Weight (g)											Length (cm)			pH	
AA (%)	AME (%)	BW	Prov ^1^	Gizzard	Pancreas	Heart ^2^	Liver	Spleen	Bursa	Duo ^1^	Jej ^1^	Ileum	Duo ^1^	Jej ^1^	Ileum	Gizzard	Ileum
70		3596	11.8	38.4	4.44	15.1	64.2	2.63	2.35	14.0	25.6	18.4	29.7	73.1	73.4	2.63	6.49
80		3871	10.6	41.4	4.77	16.3	69.9	2.71	2.48	13.5	22.6	17.4	30.6	72.4	73.0	3.01	6.54
90		3926	11.4	40.2	4.92	16.2	65.8	2.91	2.77	13.7	23.5	18.2	30.2	71.0	69.8	3.13	6.50
100		4046	10.6	40.6	5.16	15.9	73.4	2.86	2.32	14.9	24.7	17.9	32.7	73.0	71.7	3.25	6.36
SEM ^3^		107	0.56	1.55	0.22	0.64	3.03	0.26	0.26	0.74	1.25	1.05	1.01	2.52	2.56	0.11	0.15
	84	3826	10.6	37.6 ^b^	4.85	15.1	67.8	2.75	2.24	13.0 ^b^	23.4	17.1	29.8	72.3	70.8	3.04	6.33
	92	3853	11.3	40.4 ^ab^	4.88	16.0	70.9	2.59	2.54	13.7 ^ab^	23.9	19.0	30.7	70.6	71.3	3.08	6.60
	100	3899	11.4	42.5 ^a^	4.74	16.6	66.3	3.00	2.68	15.3 ^a^	25.0	17.8	31.9	74.1	73.9	2.90	6.48
	SEM	92.9	0.48	1.34	0.19	0.55	2.62	0.23	0.23	0.64	1.08	0.91	0.88	2.18	2.22	0.09	0.13
70	84					15.3 ^a^										3.06 ^abc^	
70	92					13.7 ^a^										2.51 ^bc^	
70	100					16.4 ^a^										2.33 ^c^	
80	84					16.2 ^a^										2.98 ^abc^	
80	92					17.2 ^a^										3.10 ^abc^	
80	100					15.6 ^a^										2.96 ^abc^	
90	84					13.0 ^a^										2.86 ^abc^	
90	92					17.5 ^a^										3.57 ^a^	
90	100					18.0 ^a^										2.96 ^abc^	
100	84					16.1 ^a^										3.25 ^abc^	
100	92					15.5 ^a^										3.13 ^abc^	
100	100					16.3 ^a^										3.36 ^ab^	
SEM						1.11										0.19	
*p*-value																	
AA		0.33	0.30	0.64	0.11	0.63	0.51	0.94	0.61	0.71	0.23	0.55	0.29	0.84	0.33	0.01	0.88
AME		0.39	0.42	0.04	0.82	0.19	0.17	0.43	0.44	0.03	0.27	0.08	0.21	0.51	0.23	0.40	0.88
AA × AME		0.08	0.09	0.50	0.40	0.04	0.15	0.29	0.29	0.99	0.16	0.08	0.76	0.47	0.75	0.03	0.38

^a–c^ Means in a column not sharing a common superscript were different (*p* < 0.05). ^1^ Prov stands for proventriculus, duo stands for duodenum, and Jej stands for jejunum. ^2^ Tukey’s test was not able to separate treatment means of heart weight. ^3^ SEM = standard error of mean.

**Table 7 animals-15-01064-t007:** Absolute weight and relative weight of processed meat of male and female Cobb 700 broilers at 62 d of age.

Treatment		Absolute Weight (g)								Relative Weight (%)						
AA (%)	AME (%)	BW	Carcass	Wing	Breast	Tender	Drumstick	Thighs ^1^	Fat Pad	Carcass	Wing	Breast	Tender	Drumstick	Thighs	Fat Pad
70		3628	2661	295	702	151	363	496	79.9 ^a^	73.5 ^b^	8.17	19.3 ^c^	4.20 ^b^	9.99 ^a^	13.5 ^a^	2.20^a^
80		3670	2767	307	791	168	357	488	71.1 ^ab^	75.5 ^a^	8.49	21.5 ^b^	4.61 ^a^	9.79 ^ab^	13.3 ^ab^	1.99^a^
90		3794	2859	316	846	174	363	492	61.5 ^bc^	75.4 ^a^	8.39	22.2 ^ab^	4.59 ^a^	9.57 ^b^	13.1 ^b^	1.63^b^
100		3857	2891	319	877	179	379	498	55.8 ^c^	76.2 ^a^	8.58	22.8 ^a^	4.72 ^a^	9.85 ^ab^	13.0 ^b^	1.46^b^
SEM ^2^		77.00	57.70	5.14	22.10	3.85	6.54	10.00	2.70	0.29	0.07	0.24	0.05	0.09	0.11	0.06
	84	3723	2816	309	831	170	361	489	58.8 ^b^	76.0 ^a^	8.56	22.3 ^a^	4.62 ^a^	9.74	13.2	1.62^c^
	92	3636	2739	304	792	166	356	485	64.6 ^b^	75.3 ^a^	8.40	21.7 ^a^	4.59 ^a^	9.79	13.2	1.78^b^
	100	3853	2829	315	790	168	380	506	77.8 ^a^	74.2 ^b^	8.26	20.3 ^b^	4.38 ^b^	9.88	13.2	2.06^a^
	SEM	66.70	49.90	4.45	19.10	3.33	5.66	8.67	2.34	0.25	0.06	0.21	0.04	0.07	0.09	0.05
70	84	3786 ^abc^	2837 ^ab^	312 ^ab^	784 ^abcd^	165 ^abcd^	371 ^ab^	504 ^a^			8.27 ^cd^					
70	92	3558 ^abc^	2605 ^ab^	288 ^ab^	703 ^cd^	151 ^cd^	354 ^ab^	501 ^a^			8.16 ^cd^					
70	100	3542 ^abc^	2542 ^b^	285 ^b^	618^d^	139 ^d^	363 ^ab^	483 ^a^			8.10 ^cd^					
80	84	3696 ^abc^	2805 ^ab^	308 ^ab^	827 ^abc^	176 ^abc^	361 ^ab^	493 ^a^			8.68 ^abc^					
80	92	3619 ^abc^	2741 ^ab^	301 ^ab^	790 ^abcd^	167 ^abcd^	350 ^ab^	478 ^a^			8.34 ^abcd^					
80	100	3695 ^abc^	2756 ^ab^	312 ^ab^	756 ^bcd^	161 ^abcd^	361 ^ab^	491 ^a^			8.46 ^abcd^					
90	84	3362 ^c^	2588 ^ab^	296 ^ab^	759 ^bcd^	155 ^bcd^	325 ^b^	448 ^a^			8.85 ^a^					
90	92	3885 ^abc^	2930 ^ab^	321 ^ab^	879 ^abc^	181 ^abc^	369 ^ab^	510 ^a^			8.28 ^bcd^					
90	100	4134 ^a^	3059 ^a^	331 ^a^	902 ^ab^	187 ^a^	396 ^a^	519 ^a^			8.04^d^					
100	84	4048 ^ab^	3034 ^ab^	321 ^ab^	952 ^a^	186 ^ab^	389 ^a^	512 ^a^			8.46 ^abcd^					
100	92	3483 ^bc^	2678 ^ab^	305 ^ab^	795 ^abcd^	164 ^abcd^	350 ^ab^	450 ^a^			8.84 ^ab^					
100	100	4042 ^ab^	2961 ^ab^	331 ^a^	883 ^abc^	186 ^ab^	399 ^a^	531 ^a^			8.44 ^abcd^					
SEM		133.40	99.90	8.90	38.20	6.66	11.32	17.30			0.12					
*p*-value																
AA		0.14	0.03	0.01	<0.01	<0.01	0.11	0.89	<0.01	<0.01	<0.01	<0.01	<0.01	0.01	<0.01	<0.01
AME		0.08	0.38	0.21	0.24	0.62	0.01	0.21	<0.01	<0.01	<0.01	<0.01	<0.01	0.38	0.99	<0.01
AA × AME		<0.01	<0.01	0.02	<0.01	<0.01	<0.01	0.01	0.43	0.43	<0.01	0.09	0.08	0.12	0.37	0.06

^a–d^ Means in a column not sharing a common superscript were different (*p <* 0.05). ^1^ Tukey’s test was not able to separate treatments means of thighs weight. ^2^ SEM = standard error of mean.

**Table 8 animals-15-01064-t008:** Woody breast myopathy (WBM) incidence (%) of Cobb 700 broilers at 33, 40, 47, and 62 days of age.

Treatment		WBM ^2^ Day 33			WBM ^2^ Day 40				WBM ^2^ Day 47				WBM ^2^ Day 62			
AA (%)	AME (%)	0	1	2	0	1	2	3	0	1	2	3	0	1	2	3
70		99.5 ^a^	0.46 ^b^	0.00	96.3 ^a^	3.24 ^b^	0.46	0.00	86.6 ^a^	9.72	3.24 ^b^	0.46 ^b^	82.0 ^a^	15.2 ^b^	2.78	0.00 ^b^
80		98.6 ^ab^	0.93 ^b^	0.46	83.3 ^b^	15.28 ^a^	0.93	0.46	69.0 ^b^	17.13	8.80 ^ab^	5.09 ^ab^	64.3 ^b^	22.8 ^ab^	12.04	0.93 ^b^
90		94.0 ^bc^	4.17 ^ab^	1.85	84.3 ^b^	12.04 ^a^	2.31	1.39	67.6 ^b^	17.59	11.11 ^a^	3.70 ^ab^	61.1 ^b^	25.0 ^ab^	8.33	5.56 ^ab^
100		93.5^c^	5.56 ^a^	0.93	75.9 ^b^	17.59 ^a^	3.70	2.78	62.5 ^b^	16.67	11.57 ^a^	9.26 ^a^	48.3 ^b^	33.7 ^a^	9.44	8.52 ^a^
SEM ^1^		1.27	1.05	0.62	2.56	1.97	1.04	1.24	3.28	2.67	1.65	1.95	4.54	3.84	3.25	1.90
	84	96.5	2.78	0.69	81.9 ^b^	13.54	2.43	2.08	67.7 ^b^	17.71	8.68	5.90	53.5 ^b^	31.7 ^a^	9.16	5.69
	92	95.8	2.78	1.39	83.0 ^ab^	13.89	2.08	1.04	69.1 ^ab^	15.63	10.07	5.21	64.7 ^ab^	24.2 ^ab^	8.33	2.78
	100	96.9	2.78	0.35	89.9 ^a^	8.68	1.04	0.35	77.4 ^a^	12.50	7.29	2.78	73.6 ^a^	16.7 ^b^	6.94	2.78
	SEM	1.10	0.91	0.54	2.22	1.71	0.90	1.08	2.84	2.31	1.43	1.69	3.93	3.33	2.81	1.64
*p*-value																
AA		<0.01	<0.01	0.22	<0.01	<0.01	0.13	0.42	<0.01	0.12	<0.01	0.02	<0.01	0.01	0.24	0.01
AME		0.71	0.97	0.39	0.03	0.06	0.53	0.54	0.04	0.27	0.43	0.40	<0.01	0.01	0.85	0.30
AA × AME		0.34	0.66	0.27	0.77	0.25	0.41	0.31	0.21	0.07	0.47	0.88	0.86	0.43	0.96	0.35

^a–c^ Means in a column not sharing a common superscript were different (*p* < 0.05). ^1^ SEM = standard error of mean. ^2^ WBM severity categories: 0 = normal, 1 = slight, 2 = moderate, and 3 = severe.

**Table 9 animals-15-01064-t009:** Footpad dermatitis length (cm) and the average number of broilers with footpad dermatitis at 47 days, the footpad dermatitis length (cm) of normal broilers and broilers with woody breast myopathy (WBM) at 57 days, and marginal return at 62 days of age in Cobb 700 broilers.

Treatment		Footpad Dermatitis D 47						Footpad Dermatitis D 57		Marginal Return D 62		
AA (%)	AME (%)	Female Len ^2^	Male Len	Len	Female Number	Male Number	Total Number	Normal Len	WBM Len	WBM Was not Included	WBM 3 Was Included	WBM 2 and 3 Were Included
70		0.29 ^b^	0.25 ^b^	0.37 ^b^	0.33 ^c^	0.44 ^b^	0.78 ^c^	0.14 ^b^	0.14	1.58 ^b^	1.58 ^a^	1.56
80		0.44 ^ab^	0.33 ^b^	0.67 ^ab^	1.06 ^bc^	0.89 ^b^	1.94 ^bc^	0.20 ^ab^	0.18	1.76 ^ab^	1.74 ^a^	1.61
90		0.86 ^a^	0.56 ^ab^	0.85 ^a^	1.89 ^ab^	1.61 ^ab^	3.50 ^ab^	0.34 ^ab^	0.36	1.87 ^a^	1.80 ^a^	1.75
100		0.76 ^a^	0.91 ^a^	0.91 ^a^	2.22 ^a^	2.22 ^a^	4.44 ^a^	0.59 ^a^	0.88	1.83 ^a^	1.81 ^a^	1.70
SEM ^1^		0.11	0.14	0.12	0.29	0.32	0.51	0.12	0.15	0.06	0.06	0.07
	84	0.78 ^a^	0.75	0.88 ^a^	1.92 ^a^	2.08 ^a^	4.00 ^a^	0.48 ^a^	0.51	1.87 ^a^	1.84	1.74
	92	0.34 ^b^	0.36	0.49 ^b^	0.92 ^b^	1.00 ^b^	1.92 ^b^	0.13 ^b^	0.17	1.75 ^ab^	1.72	1.65
	100	0.63 ^ab^	0.43	0.73 ^ab^	1.29 ^ab^	0.79 ^b^	2.08 ^b^	0.34 ^ab^	0.50	1.65 ^b^	1.63	1.57
	SEM	0.10	0.12	0.11	0.25	0.28	0.44	0.10	0.13	0.05	0.05	0.06
*p*-value												
AA		<0.01	0.01	0.01	<0.01	<0.01	<0.01	0.04	0.05	0.01	0.05	0.29
AME		0.01	0.05	0.03	0.02	<0.01	<0.01	0.05	0.27	0.03	0.05	0.16
AA × AME		0.97	0.92	0.95	0.31	0.93	0.70	0.24	0.80	0.27	0.25	0.35

^a–c^ Means in a column not sharing a common superscript were different (*p* < 0.05). ^1^ SEM = standard error of mean. ^2^ Len stands for the footpad dermatitis length.

**Table 10 animals-15-01064-t010:** Internal organ weights, intestine length, and pH of Ross 708 broilers without woody breast at 58 d of age.

Treatment		Weight (g)											Length (cm)			pH	
AA (%)	AME (%)	BW	Prov ^1^	Gizzard	Pancreas	Heart	Liver	Spleen	Bursa	Duo ^1^	Jej ^1,2^	Ileum ^2^	Duo ^1^	Jej ^1,2^	Ileum ^2^	Gizzard	Ileum
70		3265	11.83 ^a^	39.3	4.88	14.2	63.3	3.19 ^ab^	3.45 ^a^	14.7	27.3 ^a^	20.4 ^a^	30.6	73.8 ^a^	73.8 ^a^	2.52 ^b^	6.51
80		3702	10.48 ^ab^	37.3	4.68	14.2	67.9	3.37 ^a^	2.66 ^ab^	14.7	26.3 ^a^	18.8 ^a^	30.8	75.8 ^a^	73.5 ^a^	3.06 ^a^	6.58
90		3821	10.22 ^ab^	37.7	4.98	15.1	64.4	3.36 ^a^	2.82 ^ab^	13.1	23.8 ^a^	17.8 ^a^	29.3	69.5 ^a^	68.5 ^a^	3.11 ^a^	6.45
100		3594	9.41 ^b^	37.6	4.81	14.5	59.4	2.43 ^b^	1.82 ^b^	14.0	23.2 ^a^	17.0 ^a^	30.6	69.4 ^a^	67.6 ^a^	3.35 ^a^	6.31
SEM ^3^		114.00	0.63	1.59	0.24	0.61	0.05	0.23	0.30	0.69	1.24	0.97	0.70	2.17	2.56	0.14	0.14
	84	3710	10.27	37.0	4.94	14.5	68.3	2.96 ^ab^	2.26	13.8	23.3 ^b^	17.8	29.7	71.1	69.6	3.08	6.44
	92	3472	10.90	37.3	4.98	13.8	60.8	2.71 ^b^	2.90	13.6	25.0 ^ab^	18.6	30.2	73.2	70.8	3.10	6.41
	100	3605	10.28	39.7	4.59	15.2	62.1	3.60 ^a^	2.91	15.0	27.2 ^a^	19.0	31.0	72.1	72.2	2.85	6.54
	SEM	98.90	0.55	1.38	0.21	0.53	0.05	0.20	0.26	0.60	1.07	0.84	0.60	1.88	2.22	0.12	0.12
70	84	3631 ^ab^															
70	92	3252 ^ab^															
70	100	2911 ^b^															
80	84	3901 ^a^															
80	92	3540 ^ab^															
80	100	3666 ^ab^															
90	84	3794 ^ab^															
90	92	3690 ^ab^															
90	100	3979 ^a^															
100	84	3514 ^ab^															
100	92	3406 ^ab^															
100	100	3863 ^ab^															
SEM		198.00															
*p*-value																	
AA		0.20	0.02	0.31	0.83	0.76	0.08	0.01	0.01	0.05	0.01	0.01	0.06	0.03	0.02	0.01	0.73
AME		0.75	0.43	0.29	0.31	0.40	0.31	0.01	0.14	0.24	0.03	0.41	0.23	0.51	0.53	0.24	0.64
AA × AME		0.03	0.81	0.86	0.06	0.58	0.07	0.35	0.35	0.06	0.58	0.06	0.14	0.34	0.09	0.06	0.33

^a, b^ Means in a column not sharing a common superscript were different (*p* < 0.05). ^1^ Prov stands for proventriculus, duo stands for duodenum, and Jej stands for jejunum. ^2^ Tukey’s test was not able to separate treatment means of the weight and length of jejunum and ileum. ^3^ SEM = standard error of mean.

**Table 11 animals-15-01064-t011:** Internal organ weights, intestine length, and pH of Ross 708 broilers with woody breast myopathy (WBM) at 59 d of age.

Treatment		Weight (g)											Length (cm)			pH	
AA (%)	AME (%)	BW	Prov ^1^	Gizzard	Pancreas	Heart	Liver	Spleen	Bursa	Duo ^1^	Jej ^1^	Ileum	Duo ^1^	Jej ^1^	Ileum	Gizzard	Ileum
80		4031	10.86	38.2	5.09	16.1	75.5	2.74	2.02	13.9	25.1	68.9	29.6	70.7	17.7	3.00	6.57
90		4030	9.36	36.8	5.26	15.5	70.7	3.57	2.59	13.9	25.1	67.3	28.4	68.5	18.5	3.05	6.58
100		3892	9.18	39.4	4.97	14.5	68.3	2.93	2.23	13.6	24.1	66.5	29.4	67.6	18.4	3.33	6.61
SEM ^2^		137.00	0.60	1.51	0.30	0.80	4.32	0.26	0.27	0.72	1.37	0.97	0.64	2.42	2.43	0.11	0.11
	84	4001	9.09	35.9 ^b^	5.18	15.3	73.6	2.93	1.97	13.5	22.9 ^b^	67.7	28.8	72.2	17.8	3.27	6.71
	92	3896	10.31	36.9 ^ab^	4.97	14.5	69.0	2.74	2.24	12.9	23.3 ^ab^	66.2	28.9	65.5	17.4	3.15	6.54
	100	4057	9.99	41.6 ^a^	5.18	16.3	71.9	3.57	2.62	15.0	28.1 ^a^	68.8	29.8	69.1	19.5	2.97	6.51
	SEM	119.00	0.52	1.30	0.26	0.69	3.74	0.22	0.24	0.62	1.18	0.84	0.55	2.09	2.10	0.10	0.10
*p*-value																	
AA		0.65	0.10	0.48	0.84	0.40	0.48	0.09	0.32	0.97	0.81	0.78	0.48	0.67	0.83	0.11	0.99
AME		0.77	0.22	0.05	0.83	0.36	0.80	0.11	0.23	0.15	0.03	0.78	0.51	0.11	0.35	0.26	0.55
AA × AME		0.28	0.05	0.27	0.84	0.69	0.49	0.64	0.73	0.40	0.56	0.36	0.34	0.86	0.41	0.17	0.71

^a, b^ Means in a column not sharing a common superscript were different (*p* < 0.05). ^1^ Prov stands for proventriculus, duo stands for duodenum, and Jej stands for jejunum. ^2^ SEM = standard error of mean.

**Table 12 animals-15-01064-t012:** Absolute weight and relative weight of processing of female and male Ross 708 broilers at 64 d of age.

Treatment		Absolute Weight (g)								Relative Weight (%)						
AA (%)	AME (%)	BW	Carcass	Wing	Breast	Tender	Drumstick	Thighs	Fat Pad	Carcass	Wing	Breast	Tender	Drumstick	Thighs	Fat Pad
70		3556 ^b^	2585 ^b^	298 ^b^	656 ^c^	147	366	494	70.2 ^a^	72.6	8.43	18.3	4.06	10.32 ^a^	13.9	1.98 ^a^
80		3824 ^ab^	2917 ^a^	323 ^a^	855 ^b^	172	377	515	64.5 ^a^	76.3	8.53	22.2	4.53	9.89 ^b^	13.5	1.67 ^b^
90		3997 ^a^	3066 ^a^	332 ^a^	964 ^a^	181	388	531	61.6 ^a^	76.9	8.42	24.0	4.59	9.74 ^b^	13.2	1.54 ^b^
100		3842 ^ab^	2936 ^a^	324 ^a^	924 ^ab^	175	374	504	44.2 ^b^	77.1	8.49	23.9	4.56	9.75 ^b^	13.1	1.17 ^c^
SEM ^1^		117.70	93.90	8.13	34.30	8.22	11.04	17.40	6.04	0.63	0.16	0.45	0.17	0.16	0.26	0.14
	84	3874	2952	325	920 ^a^	177	378	517	53.5 ^b^	76.6	8.46	23.5	4.57	9.75 ^b^	13.3	1.39^c^
	92	3814	2901	321	850 ^ab^	169	380	510	59.0 ^b^	75.9	8.49	22.1	4.42	10.00 ^a^	13.4	1.55 ^b^
	100	3726	2775	312	780 ^b^	161	372	506	67.8 ^a^	74.6	8.45	20.6	4.32	10.02 ^a^	13.6	1.82 ^a^
	SEM	77.30	60.00	4.77	21.50	4.02	7.22	11.30	2.29	0.27	0.08	0.17	0.06	0.06	0.09	0.04
70	84					172 ^ab^				74.6 ^bc^		20.5 ^f^	4.39 ^ab^		13.9 ^ab^	
70	92					146 ^bc^				73.4 ^c^		18.4 ^g^	4.10 ^bc^		14.2 ^a^	
70	100					122 ^c^				69.6 ^d^		15.9 ^h^	3.70 ^c^		13.5 ^abc^	
80	84					173 ^ab^				77.0 ^ab^		23.8 ^abcd^	4.64 ^ab^		13.1 ^bc^	
80	92					172 ^ab^				76.5 ^ab^		22.2 ^de^	4.47 ^ab^		13.4 ^bc^	
80	100					172 ^ab^				75.5 ^abc^		20.7 ^ef^	4.47 ^ab^		13.9 ^ab^	
90	84					185 ^a^				77.1 ^ab^		24.8 ^ab^	4.69 ^a^		13.1 ^bc^	
90	92					187 ^a^				77.1 ^ab^		23.9 ^abc^	4.67 ^a^		13.1 ^bc^	
90	100					173 ^ab^				76.5 ^ab^		23.2 ^bcd^	4.40 ^ab^		13.5 ^abc^	
100	84					178 ^ab^				77.7 ^a^		25.0 ^a^	4.55 ^ab^		13.0 ^c^	
100	92					169 ^ab^				76.7 ^ab^		24.0 ^abc^	4.44 ^ab^		12.8 ^c^	
100	100					177 ^ab^				76.8 ^ab^		22.7 ^cd^	4.70 ^a^		13.4 ^bc^	
SEM						8.04				0.54		0.34	0.11		0.17	
*p*-value																
AA		0.01	<0.01	<0.01	<0.01	<0.01	0.31	0.23	<0.01	<0.01	0.80	<0.01	<0.01	<0.01	<0.01	<0.01
AME		0.40	0.11	0.18	<0.01	0.02	0.71	0.78	<0.01	<0.01	0.95	<0.01	0.01	<0.01	0.08	<0.01
AA × AME		0.64	0.35	0.32	0.55	0.04	0.32	0.24	0.18	<0.01	0.90	<0.01	0.02	0.71	0.01	0.32

^a–h^ Means in a column not sharing a common superscript were different (*p* < 0.05). ^1^ SEM = standard error of mean.

**Table 13 animals-15-01064-t013:** Woody breast myopathy (WBM) incidence (%) of Ross 708 broilers at 33, 40, 47, and 62 days of age.

Treatment		WBM^2^ Day 33			WBM Day 40				WBM Day 47				WBM Day 64			
AA (%)	AME (%)	0	1	2	0	1	2	3	0	1	2	3	0	1	2	3
70		100.00	0.00	0.00	99.50 ^a^	0.46 ^c^	0.00 ^c^	0.00 ^a^	95.80 ^a^	3.70 ^b^	0.46 ^c^	0.00 ^c^	91.50 ^a^	6.67 ^b^	1.85 ^c^	0.00
80		95.40	4.63	0.00	87.00 ^b^	9.72 ^b^	2.31 ^bc^	0.93 ^a^	68.50 ^b^	19.40 ^a^	8.33 ^b^	3.70 ^bc^	72.20 ^b^	13.52 ^b^	11.48 ^b^	2.78
90		90.70	8.80	0.46	73.60 ^c^	22.22 ^a^	4.17 ^ab^	0.00 ^a^	52.80 ^c^	21.30 ^a^	15.74 ^a^	10.19 ^ab^	58.30 ^b^	27.59 ^a^	13.15 ^ab^	0.93
100		90.70	7.87	1.39	67.60 ^c^	23.61 ^a^	6.48 ^a^	2.31 ^a^	47.20 ^c^	25.50 ^a^	15.28 ^a^	12.04 ^a^	58.50 ^b^	17.04 ^ab^	21.48 ^a^	2.96
SEM ^1^		1.29	1.31	0.45	2.55	2.07	1.09	0.67	2.97	2.88	1.62	1.73	4.10	3.71	2.52	1.01
	84	89.9	9.38	0.69	74.0 ^b^	20.83 ^a^	4.51	0.69	57.60 ^b^	17.70	15.63 ^a^	9.03	58.50 ^b^	19.86	17.36 ^a^	4.31^a^
	92	95.5	3.82	0.69	83.7 ^a^	11.81 ^b^	3.13	1.39	70.80 ^a^	15.60	8.68 ^b^	4.86	72.20 ^a^	14.72	13.06 ^a^	0.00^b^
	100	97.2	2.78	0.00	88.2 ^a^	9.38 ^b^	2.08	0.35	69.80 ^a^	19.10	5.56 ^b^	5.56	79.70 ^a^	14.03	5.56 ^b^	0.69^b^
	SEM	1.12	1.13	0.39	2.21	1.79	0.94	0.58	2.57	2.49	1.40	1.50	3.55	3.22	2.18	0.88
70	84	100.00 ^a^	0.00 ^c^													
70	92	100.00 ^a^	0.00 ^c^													
70	100	100.00 ^a^	0.00 ^c^													
80	84	88.90 ^bc^	11.11 ^ab^													
80	92	98.60 ^ab^	1.39 ^bc^													
80	100	98.60 ^ab^	1.39 ^bc^													
90	84	81.90 ^c^	16.67 ^a^													
90	92	91.70 ^abc^	8.33 ^abc^													
90	100	98.60 ^ab^	1.39 ^bc^													
100	84	88.90 ^bc^	9.72 ^abc^													
100	92	91.70 ^abc^	5.56 ^bc^													
100	100	91.70 ^abc^	8.33 ^abc^													
SEM		2.24	2.26													
*p*-value																
AA		<0.01	<0.01	0.08	<0.01	<0.01	<0.01	0.03	<0.01	<0.01	<0.01	<0.01	<0.01	<0.01	<0.01	0.10
AME		<0.01	<0.01	0.49	<0.01	<0.01	0.20	0.57	<0.01	0.66	<0.01	0.13	<0.01	0.34	<0.01	<0.01
AA × AME		0.01	0.01	0.54	0.09	0.11	0.69	0.71	0.37	0.69	0.18	0.78	0.31	0.61	0.71	0.16

^a–c^ Means in a column not sharing a common superscript were different (*p* < 0.05). ^1^ SEM = standard error of mean. ^2^ WBM severity categories: 0 = normal, 1 = slight, 2 = moderate, and 3 = severe.

**Table 14 animals-15-01064-t014:** Footpad dermatitis length (cm) and the average number of broilers with footpad dermatitis at 47 days, the footpad dermatitis length (cm) of normal broilers and broilers with woody breast myopathy (WBM) at 59 days, and marginal return at 64 days of age in Ross 708 broilers.

Treatment			Footpad Dermatitis D47					Footpad Dermatitis D59		Marginal Return D64		
AA (%)	AME (%)	Female Len	Male Len	Len	Female Number	Male Number	Total Number	Normal Birds	Birds with WBM	WBM Was not Considered	WBM 3 Was Considered	WBM 2 and 3 Were Considered
70		0.07 ^c^	0.00 ^c^	0.07 ^c^	0.06	0.00	0.06	0.06 ^b^		1.45 ^c^	1.45 ^c^	1.45 ^c^
80		0.29 ^bc^	0.14 ^bc^	0.38 ^bc^	0.44	0.50	0.94	0.03 ^b^	0.06 ^a^	1.94 ^b^	1.91 ^b^	1.81 ^b^
90		0.42 ^ab^	0.36 ^b^	0.53 ^ab^	1.39	1.06	2.44	0.21 ^ab^	0.55 ^a^	2.32 ^a^	2.32 ^a^	2.24 ^a^
100		0.64 ^a^	0.82 ^a^	0.83 ^a^	1.67	2.00	3.67	0.48 ^a^	0.53 ^a^	1.87 ^b^	1.84 ^b^	1.64 ^bc^
SEM ^1^		0.09	0.10	0.09	0.24	0.22	0.41	0.08	0.15	0.08	0.08	0.07
	84	0.68 ^a^	0.61 ^a^	0.84 ^a^	1.79	1.88	3.67	0.46 ^a^	0.77 ^a^	2.24 ^a^	2.20 ^a^	2.08 ^a^
	92	0.32 ^b^	0.33 ^a^	0.42 ^b^	0.79	0.71	1.50	0.13 ^b^	0.28 ^ab^	1.81 ^b^	1.81 ^b^	1.71 ^b^
	100	0.08 ^b^	0.05 ^b^	0.10 ^c^	0.08	0.08	0.17	0.00 ^b^	0.09 ^b^	1.62 ^b^	1.62 ^b^	1.58 ^b^
	SEM	0.08	0.08	0.08	0.20	0.19	0.35	0.07	0.13	0.07	0.07	0.07
70	84				0.17 ^c^	0.00 ^c^	0.17 ^c^					
70	92				0.00 ^c^	0.00 ^c^	0.00 ^c^					
70	100				0.00 ^c^	0.00 ^c^	0.00 ^c^					
80	84				1.00 ^bc^	1.17 ^bc^	2.17 ^bc^					
80	92				0.33 ^c^	0.33 ^bc^	0.67 ^c^					
80	100				0.00 ^c^	0.00 ^c^	0.00 ^c^					
90	84				2.67 ^ab^	2.17 ^b^	4.83 ^ab^					
90	92				1.33 ^bc^	1.00 ^bc^	2.33 ^bc^					
90	100				0.17 ^c^	0.00 ^c^	0.17 ^c^					
100	84				3.33 ^a^	4.17 ^a^	7.50 ^a^					
100	92				1.50 ^abc^	1.50 ^bc^	3.00 ^bc^					
100	100				0.17 ^c^	0.33 ^bc^	0.50 ^c^					
SEM					0.41	0.39	0.71					
*p*-value												
AA		<0.01	<0.01	<0.01	<0.01	<0.01	<0.01	0.01	0.04	<0.01	<0.01	0.00
AME		<0.01	<0.01	<0.01	<0.01	<0.01	<0.01	<0.01	0.02	<0.01	<0.01	0.00
AA × AME		0.37	0.05	0.05	0.02	<0.01	<0.01	0.15	0.20	0.60	0.38	0.07

^a–c^ Means in a column not sharing a common superscript were different (*p* < 0.05). ^1^ SEM = standard error of mean. ^2^ Len stands for the footpad dermatitis length.

## Data Availability

The raw data supporting the conclusions of this article will be made available by the authors on request.

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
