# Peer review of "Effects of Reduced Amino Acids and Apparent Metabolizable Energy on Meat Processing, Internal Organ Development, and Economic Returns of Cobb 700 and Ross 708 Broilers"

_animals, 2025, doi:10.3390/ani15071064_

Round 1

Reviewer 1 Report

Comments and Suggestions for Authors
  1. Provide more physiological context in the introduction regarding how AA and AME reductions affect metabolic pathways in broilers.
  2. Clearly define the rationale for selecting the specific levels of AA and AME reductions.
  3. methodology is well-documented and replicable, covering key parameters such as:
    • Processing and internal organ development assessments
    • Woody breast myopathy evaluation via manual palpation
    • Economic return calculations considering WBM incidence. However, some few clarifications are needed; (i). Were all birds assessed for WBM, or only a subsample?(ii) Were gut microbiota differences between diet groups considered?
  4. Explain why Tukey-Kramer was the preferred post hoc test.
  5. Discuss the practical applicability of these findings for the poultry industry, particularly regarding cost-effective feeding strategies

Author Response

Dear Editor and Reviewers,

We sincerely appreciate your thoughtful and constructive feedback on our manuscript. We have carefully addressed each of your comments and made the necessary revisions accordingly.

Your insights are invaluable to us, and we are committed to ensuring that our manuscript meets the highest standards. If you have any further suggestions or believe additional improvements are needed, we would be happy to make further revisions.

Thank you for your time and consideration. We look forward to your feedback.

Commens 1:  Provide more physiological context in the introduction regarding how AA and AME reductions affect metabolic pathways in broilers.

Response 1: The following information has been added to the Introduction:

The reduction of dietary AA and AME affects broiler growth by modulating blood metabolites related to protein and lipid metabolism. The reduction in dietary AME limits the broiler’s capacity to efficiently utilize excess protein, leading to its accumulation in the intestines. This not only results in nutrient wastage but also promotes the proliferation of harmful intestinal bacteria [1], ultimately impairing gut health. Conversely, dietary protein deficiency or an imbalanced AA profile can slow growth rates, increase fat deposition, and damage the intestinal mucosa and barrier function [2]. These impairments reduce nutrient digestion and absorption, thereby negatively impacting broiler performance.

An excessively low protein-to-energy ratio results in disproportionately high energy intake relative to protein, which has been shown to significantly elevate plasma triglyceride concentrations in broilers [3,4]. This imbalance stimulates hepatic lipogenesis, as indicated by increased activities of key lipogenic enzymes, including fatty acid synthase, malate dehydrogenase, and acetyl-CoA carboxylase [5]. As a result, lipogenesis and fat deposition are enhanced in response to excess energy intake [6]. Furthermore, broiler growth is closely regulated by serum hormone levels. Broilers fed low-protein diets exhibit elevated plasma concentrations of growth hormone [7] and thyroid hormones [8], which together stimulate skeletal muscle development and protein synthesis [9]. These hormonal changes enhance the utilization efficiency of the limited dietary protein and reduce plasma uric acid levels, the end product of protein catabolism, thereby mitigating the physiological consequences of protein deficiency.

Commens 2:  Clearly define the rationale for selecting the specific levels of AA and AME reductions.

Response 2: One previous study found that the WBM incidence decreased at 42 d of age when dietary lysine was reduced by 30% from 12 to 28 d of age [10]. Since we would like to investigate different levels of AA reduction, we have four levels of AA, including 70%, 80%, 90%, and 100% of high breeder recommendations.

AME is another important nutrition for the broiler feed, so it is necessary to explore regarding the simultaneous reduction in both amino acids (AAs) and energy in broiler diets. We try to reduce the broiler feed’s AME as much as possible, and 84% is lowest level of AME that could be reduced based on the current corn-soybean feed. We also would like to investigate different levels of AME reduction, so we have 92% AME, which corresponds to the half 16% for the 84% AME level.

Commens 3:  Methodology is well-documented and replicable, covering key parameters such as:

Processing and internal organ development assessments, Woody breast myopathy evaluation via manual palpation, Economic return calculations considering WBM incidence.

However, some few clarifications are needed; (i). Were all birds assessed for WBM, or only a subsample? (ii) Were gut microbiota differences between diet groups considered?

Response 3: (1) Yes, all birds in each pen were assess for WBM.

(2)  The gut microbiota of different diets is a very interesting research topic, but we have not considered the gut microbiota in this study.

Commens 4:  Explain why Tukey-Kramer was the preferred post hoc test.

Response 4: After significant ANOVA result, we need to determine which specific groups or treatment means are significantly different among multiple comparisons due to the two factorial (4 AA × 3 AME) trial design by using Tukey-Kramer test. The reason of using Tukey-Kramer method is that this method could properly control type I error rate caused by the multiple comparisons.

Commens 5:  Discuss the practical applicability of these findings for the poultry industry, particularly regarding cost-effective feeding strategies

Response 5: The discussion of economic returns has been improved as follows:

The AA and AME interaction results show that 90%AA-84%AME-fed Ross 708 and Cobb 700 broilers have the best marginal return, and the feed protein-energy ratio was both increased in these two treatments. This finding suggests that lowering the relative level of dietary AME, thereby increasing the relative AA content, can enhance the final economic return by improving the yield of high-value chicken cut-up parts. Compared to diets with relatively higher AME, which tend to promote excess energy deposition as fat, this approach offers greater value to the broiler industry through improved carcass composition and profitability. Therefore, we recommend feeding AME: 12.70 MJ/kg, CP: 25.12% (PER: 19.78 g/MJ) for Starter (0-10 d); AME: 13.00 MJ/kg, CP: 22.77% (PER: 17.51 g/MJ) for Grower (11-24 d); AME: 13.39 MJ/kg, CP: 21.46% (PER: 16.03 g/MJ) for Finisher (25-39 d); AME: 13.49 MJ/kg, CP: 20.57% (PER: 15.25 g/MJ) for Withdrawal (40-63 d) to improve economic returns for broiler companies.

  1. Apajalahti, J.; Vienola, K. Interaction between chicken intestinal microbiota and protein digestion. Animal Feed Science and Technology 2016, 221, 323-330.
  2. Barekatain, R.; Nattrass, G.; Tilbrook, A.; Chousalkar, K.; Gilani, S. Reduced protein diet and amino acid concentration alter intestinal barrier function and performance of broiler chickens with or without synthetic glucocorticoid. Poult. Sci. 2019, 98, 3662-3675.
  3. Yang, H.-M.; Wei, W.; Wang, Z.-Y.; Zhi, Y.; Yan, W.; Hou, B.-H.; Huang, K.-H.; Hao, L. Effects of early energy and protein restriction on growth performance, clinical blood parameters, carcass yield, and tibia parameters of broilers. Journal of integrative agriculture 2016, 15, 1825-1832.
  4. Malheiros, R.D.; Moraes, V.M.; Collin, A.; Janssens, G.P.; Decuypere, E.; Buyse, J. Dietary macronutrients, endocrine functioning and intermediary metabolism in broiler chickens: Pair wise substitutions between protein, fat and carbohydrate. Nutr. Res. 2003, 23, 567-578.
  5. Jariyahatthakij, P.; Chomtee, B.; Poeikhampha, T.; Loongyai, W.; Bunchasak, C. Effects of adding methionine in low-protein diet and subsequently fed low-energy diet on productive performance, blood chemical profile, and lipid metabolism-related gene expression of broiler chickens. Poult. Sci. 2018, 97, 2021-2033.
  6. Fouad, A.; El-Senousey, H. Nutritional factors affecting abdominal fat deposition in poultry: a review. Asian-australas. J. Anim. Sci. 2014, 27, 1057.
  7. Caperna, T.; Rosebrough, R.; McMurtry, J.; Vasilatos-Younken, R. Influence of dietary protein on insulin-like growth factor binding proteins in the chicken. Comparative Biochemistry and Physiology Part B: Biochemistry and Molecular Biology 1999, 124, 417-421.
  8. Buyse, J.; Decuypere, E.; Berghman, L.; Kuhn, E.; Vandesande, F. Effect of dietary protein content on episodic growth hormone secretion and on heat production of male broiler chickens. Br. Poult. Sci. 1992, 33, 1101-1109.
  9. Suthama, N.; HAYASHI, K.; Toyomizu, M.; Tomita, Y. Effect of dietary thyroxine on growth and muscle protein metabolism in broiler chickens. Poult. Sci. 1989, 68, 1396-1401.
  10. Cruz, R.; Vieira, S.; Kindlein, L.; Kipper, M.; Cemin, H.; Rauber, S. Occurrence of white striping and wooden breast in broilers fed grower and finisher diets with increasing lysine levels. Poult. Sci. 2017, 96, 501-510.

Reviewer 2 Report

Comments and Suggestions for Authors

Minor comments attached.

Author Response

Dear Editor and Reviewers,

We sincerely appreciate your thoughtful and constructive feedback on our manuscript. We have carefully addressed each of your comments and made the necessary revisions accordingly.

Your insights are invaluable to us, and we are committed to ensuring that our manuscript meets the highest standards. If you have any further suggestions or believe additional improvements are needed, we would be happy to make further revisions.

Thank you for your time and consideration. We look forward to your feedback.

Commons 1:  Title of the manuscript can be revised as "Effects of reduced amino acids and apparent metabolizable energy on meat processing, internal organ development, and economic returns of Cobb 700 and Ross 708 broilers".

Response 1: Modified as requested.

Commons 2:  Line 77: check the word “Beyond”, which is in sentence case.

Response 2: Modified as requested. Thank you!

Commons 3:  Line 173: P value must be written in italics. Please check the journal guide and revise if needed.

Response 3: Modified as requested. Thank you!

Commons 4: Is factorial arrangement a suitable design for presenting and interpreting data on WBM-severity (Tables 4 & 9)? Check and revise if needed. Chi-Square analysis can also be used.

Response 4: The WBM severity data is a percentage type data, and Chi-Square method is one of the methods being used to determine if significant difference exist between groups. However, considering too many comparisons between any two groups, it might be reasonable to consider other suitable methods, including the method of treating WBM data as a continuous data. The normality of WBM percentage data was checked and found that the WBM percentage data generally follow normal distribution. Then, we analyzed the WBM percentage data as a continue variable.

Commons 5:  Lines 179-181 /Table 1: It is better to report the organ weight and length as relative weights. Is there any reason for "absolute weights"? Probably, you can indicate both?

Response 5: In our analysis, we actually have both absolute weights and relative weights results, and we found both results are consistent with each other. Considering that the width of the table, it is very hard to include both results into the table, therefore we only include absolute weight results. We do have considered include the relative weight results into a separate table, but we already have 10 tables in the manuscript, so we did not include the relative weight into a separated table.

Commons 6:  Indicate Table 1 (Lines 178-180) & Table 2 (Lines 182-189) within the text (before the appearance of Table).

Response 6: Modified as requested.

Commons 7:  Lines 191-192: “……compared to those fed 100% AA….”? also, 90%, I think. Check and revise if needed.

Response 7: Modified as requested.

Feeding broilers 70% AA increased broiler fat pad weight compared to those fed 90% and 100% AA (P<0.01, Table 3);

Commons 8:  Revise the table 3-heading.  The word "processing" can be replaced with "processed meat"?

Response 8: Modified as requested.

Commons 9:  Avoid starting a sentence with numbers.  Ex: lines 245, 253 & 299. 

Response 9: Modified as requested.

lines 245: Broilers fed 84% AME had a significantly lower jejunum weight (P=0.03) compared to those fed 100% AME.

lines 253: Among all AA levels, broilers fed 70% AA showed the lowest carcass weight (all P<0.01).

lines 299: Ross 708 broilers fed 84% AME achieved the highest marginal return across all three AME levels under three different conditions (all P<0.01).

Commons 10:  Section 3.2.1 - information on spleen weight is missing.

Response 10: Following information added:

For the spleen, 80% and 90% AA fed broilers increased spleen as compared with 100% AA-fed broilers (P=0.01); however, 92% AME fed broilers decreased spleen as compared with 100% AME-fed broilers (P=0.01).

Commons 11:  Line 253: is the carcass weight absolute / relative?  Sentences in lines 253-255 can be combined.  Also, is the fat pad mentioned in line 259 absolute / relative?

Response 11: Modified as requested.

Among all AA levels, broilers fed 70% AA showed the lowest absolute weights of carcass, wing and breast among broilers fed all levels of AA (all P<0.01).

Commons 12:  Similar to above comment, rewrite sections 284-287.

Response 12: Modified as requested.

At day 33, broilers fed 90% AA with 84% AME showed a lower proportion of normal breast compared to six other treatments (P<0.01) and a higher proportion of slight WBM than seven other treatments (P=0.01, Table 9), However, these differences were no longer significant after day 40.

Commons 13:  Line 290: footpad dermatitis or footpad lesion (Table 10)?  Be consistent.

Response 13: All footpad lesions have been changed to footpad dermatitis.

Commons 14:  Line 293: d 58 or d 59 (Table 10)?  Check.

Response 14: It’s day 59, Modified as requested.

Commons 15:  What is "BWM" in table 10-heading?  Remove all abbreviations in all table headings and write them in full.

Response 15: It's a typo, it’s WBM. Modified as requested.

Commons 16:  All P values must be written in italics.  Please check the journal guide and revise if needed.

Response 16: Modified as requested.

Reviewer 3 Report

Comments and Suggestions for Authors

The study investigated the effects of amino acid and energy reduction on processing, internal organ development, and economic returns of Cobb 700 and Ross 708 broilers. Comments are as follows.

  1. The word “economic return” did not fit an academic paper, as it varies with local economic situation. Meat characteristics or carcass yield are better instead. Originally, it was difficult to understand the marginal return.   
  2. Show diet formulation and AA content in a main table, not a supplemental table. 
  3. Show the representative picture of WBM for scoring. And describe how many people scored the meat.
  4. Overall, the study is a nutritional trial, but the focusing points are too wide, and it is difficult to sense the academic significance. The study is like a research note. In this sense, the manuscript needs to be more developed as an academic paper.

Author Response

Dear Editor and Reviewers,

We sincerely appreciate your thoughtful and constructive feedback on our manuscript. We have carefully addressed each of your comments and made the necessary revisions accordingly.

Your insights are invaluable to us, and we are committed to ensuring that our manuscript meets the highest standards. If you have any further suggestions or believe additional improvements are needed, we would be happy to make further revisions.

Thank you for your time and consideration. We look forward to your feedback.

Commens 1:  The word “economic return” did not fit an academic paper, as it varies with local economic situation. Meat characteristics or carcass yield are better instead. Originally, it was difficult to understand the marginal return. 

Response 1: dear reviewer, this article focuses on the broiler market in the United States; therefore, the national average price of broiler chickens in the U.S. is used, and economic conditions in other regions are not considered at this time.

To calculate economic returns, the normal breast meat price was based on the standard market value, which was $0.9459/lb, according to the USDA AMS Livestock, Poultry & Grain Market News report published on Friday, September 13, 2019. The price of breast meat exhibiting severe white striping and woody breast (WBM) was assumed to be 50% of the normal breast meat price. Prices for other chicken parts were also sourced from the same USDA AMS report.

Feed ingredient prices were obtained from Feedstuffs Ingredient Market Prices on September 22, 2017, corresponding to the time of feed formulation. The feed costs for starter, grower, finisher, and withdrawal control diets were $264.97, $261.19, $245.16, and $237.74 per ton, respectively. For amino acid (AA)-reduced diets, the feed costs were $240.97, $231.24, $225.02, and $217.71 per ton, respectively.

Commens 2:  Show diet formulation and AA content in a main table, not a supplemental table.

Response 2: Modified as requested

Commens 3:  Show the representative picture of WBM for scoring. And describe how many people scored the meat.

Response 3: We appreciate the reviewer’s suggestion. In this study, all WBM (woody breast myopathy) scoring was performed consistently by a single trained evaluator to ensure scoring uniformity and reduce inter-observer variability. However, we did not capture representative images of WBM specifically for this experiment. We acknowledge the value of visual references and will consider including representative images in future studies to improve clarity and reproducibility.

Commens 4:  Overall, the study is a nutritional trial, but the focusing points are too wide, and it is difficult to sense the academic significance. The study is like a research note. In this sense, the manuscript needs to be more developed as an academic paper.

Response 4: Thank you for your valuable feedback. We appreciate your suggestion to further develop the manuscript into a more focused academic paper. While the study is rooted in a nutritional trial, we would like to emphasize several novel contributions that we believe add academic significance and relevance to the field:

  1. Multifaceted Impact of Nutrient Reduction: This study systematically investigated the effects of reducing both dietary amino acids (AA) and energy on several key welfare and production traits in broilers, including abdominal fat pad weight, woody breast myopathy (WBM) incidence, and footpad dermatitis. This multifactorial approach provides a more comprehensive understanding of how nutritional adjustments influence broiler health and carcass characteristics.
  2. Post-6-Week Internal Organ Development: Unlike many nutritional trials that focus primarily on early growth phases, our study extended the investigation to examine how continued dietary AA and energy reduction affect internal organ development beyond 6 weeks of age, offering new insights into late-phase broiler physiology.
  3. Economic Optimization Framework: We determined the optimal level of dietary nutrient reduction that balances feed cost savings with carcass yield, providing practical guidance for cost-effective broiler production without compromising product quality.
  4. Economic Returns in the Context of WBM: Importantly, we also examined the economic returns under different nutrient reduction strategies, explicitly considering the incidence of severe WBM. This is helping producers make informed decisions based not only on growth performance but also on market value.

In response to your comment, we will revise the manuscript to strengthen the focus by clearly framing these contributions in the Introduction and Discussion sections. We will also ensure that the narrative consistently emphasizes how these findings contribute to current knowledge in poultry nutrition, health, and production economics.

Round 2

Reviewer 3 Report

Comments and Suggestions for Authors

The Authors adequately responded to the comments. There is no further comment.